# Consistent Multi-Class Classification from Multiple Unlabeled Datasets

**Zixi Wei**[1†]    **Senlin Shu**[1†]    **Yuzhou Cao**[2]    **Hongxin Wei**[3]    **Bo An**[4,2]    **Lei Feng**[2*]
[1]Chongqing University    [2]Nanyang Technological University
[3]Southern University of Science and Technology    [4]Skywork AI
{zixiwei,senlinshu}@stu.cqu.edu.cn, yuzhou002@e.ntu.edu.sg
weihx@sustech.edu.cn, boan@ntu.edu.sg, lfengqaq@gmail.com

## Abstract

Weakly supervised learning aims to construct effective predictive models from imperfectly labeled data. The recent trend of weakly supervised learning has focused on how to learn an accurate classifier from completely unlabeled data, given little supervised information such as class priors. In this paper, we consider a newly proposed weakly supervised learning problem called *multi-class classification from multiple unlabeled datasets*, where only multiple sets of unlabeled data and their class priors (i.e., the proportion of each class) are provided for training the classifier. To solve this problem, we first propose a *classifier-consistent method* (CCM) based on a probability transition function. However, CCM cannot guarantee risk consistency and lacks of purified supervision information during training. Therefore, we further propose a *risk-consistent method* (RCM) that progressively purifies supervision information during training by importance weighting. We provide comprehensive theoretical analyses for our methods to demonstrate the statistical consistency. Experimental results on multiple benchmark datasets across various settings demonstrate the superiority of our proposed methods.

## 1 Introduction

Deep learning techniques have achieved great success in a variety of application domains (LeCun et al., 2015), while they heavily rely on large-scale fully labeled training data. In real-world scenarios, collecting such *strong supervision* could be costly, because accurate manual annotations are expensive and time-consuming. Such a challenge motivated extensive research on weakly supervised learning, which aims to learn satisfactory predictive models with weak supervision. According to the different types of weak supervision, various learning problems are included in the framework of weakly supervised learning, such as semi-supervised learning (Miyato et al., 2018), noisy-label learning (Han et al., 2018), and partial-label learning (Wang et al., 2022).

In this paper, we consider another challenging weakly supervised learning problem (Tang et al., 2022) called *multi-class classification from multiple unlabeled datasets* (MCMU in short), where the goal is to learn a multi-class classifier from multiple unlabeled datasets with only *class-prior probabilities* (each probability indicates the proportion of examples belonging to each class). These class-prior probabilities are essential and are the sole supervision information used in this work, without which the MCMU problem becomes fully unsupervised. The MCMU problem could be frequently encountered in many real-world scenarios. For example, it is important to predict the demographic information of users in social networks for practical policy-making (Culotta et al., 2015). But due to the data privacy, collecting the information of individual users may be prohibitive. Fortunately, it is easier to obtain multiple unlabeled datasets, with their class priors collected from pre-existing census data. In this case, the studied MCMU problem can be naturally suited.

This MCMU problem stems from a classical problem called *learning from label proportions* (LLP) (Yu et al., 2014; Liu et al., 2019), while there are significant differences between our work and

---

[†]Equal contribution.
[*]Corresponding author.

Table 1: Comparisons of our proposed method with previous studies on relevant problem settings.

| Methods | Deal with 2+ class | Deal with 2+ sets | Theoretical guarantee | No negative training risk | Risk measure |
|---|---|---|---|---|---|
| Lu et al. (2019) | × | × | ✓ | × | Classification risk |
| Menon et al. (2015) | × | × | ✓ | × | Balanced risk |
| Lu et al. (2020) | × | × | ✓ | ✓ | Classification risk |
| Zhang et al. (2020b) | × | ✓ | ✓ | × | Balanced risk |
| Tsai & Lin (2020) | × | ✓ | × | ✓ | Proportion risk |
| Lu et al. (2021) | × | ✓ | ✓ | ✓ | Classification risk |
| Tang et al. (2022) | ✓ | ✓ | ✓ | × | Classification risk |
| Our proposed | ✓ | ✓ | ✓ | ✓ | Classification risk |

previous LLP studies. In the problem setting of our work, the unlabeled sets are sampled from distributions with different class priors, while most previous LLP studies considered unlabeled sets sampled from the same distribution (Liu et al., 2019; Tsai & Lin, 2020). Besides, most previous deep LLP methods conducted *empirical risk minimization* (ERM) (Dery et al., 2017; Tsai & Lin, 2020) at the proportion level, while we aim to conduct ERM at the instance level so that the learning consistency can be guaranteed. These differences make our studied MCMU more applicable than LLP. For example, MCMU can learn from an extremely small number of unlabeled sets, while most previous LLP methods that treat each set as a training unit cannot work well in this case.

The key challenge of MCMU is how to derive effective risk estimators for conducting ERM at the instance level. To address this challenge, Tang et al. (2022) proposed an unbiased risk estimator for MCMU so that the expected classification risk of fully supervised data can be unbiasedly estimated by given training data, which is also the sole work on MCMU to the best of our knowledge. However, this method results in an unreasonable training objective caused by the negative empirical risk. Although a partial-risk regularization term is further proposed to alleviate the negative empirical risk issue, the unbiasedness of the risk estimator is actually broken, hence the theoretical guarantee cannot hold, and thus the performance is still suboptimal.

To address the above limitations, we first propose a classifier-consistent method (CCM) based on a probability transition function. CCM can be considered as a multi-class extension of Lu et al. (2021) where they only studied the binary version of MCMU by solving a surrogate set classification problem with a probability transition function. However, CCM cannot guarantee risk consistency and fails to consider differentiating the true label of each training example, which could limit its empirical performance due to the lack of any purified supervision information. Therefore, we further propose a risk-consistent method (RCM) via importance weighting, which can progressively purify supervision information during the training process by dynamically calculating the weights to identify the true label of each training example.

The main contributions of our paper can be summarized as follows:

- We propose a *classifier-consistent* method (CCM) based on a probability transition function, which can be considered as a multi-class extension of Lu et al. (2021).

- We propose a *risk-consistent* method (RCM) via importance weighting. RCM is superior to CCM because it can hold risk consistency and progressively purify supervision information.

- We provide comprehensive theoretical analyses for our proposed methods CCM and RCM to demonstrate their theoretical guarantees.

- We conduct extensive experiments on benchmark datasets with various settings. Experimental results demonstrate that CCM works well but RCM consistently outperforms CCM.

## 2 RELATED STUDIES ON RELEVANT PROBLEM SETTINGS

In this section, we introduce necessary notations, related studies, and the problem setting of our work.

## 2.1 Multi-Class Classification from Fully Labeled Data

Let us begin with introducing the ordinary multi-class classification setting. Let the feature space be $\mathcal{X} \subseteq \mathbb{R}^d$ and the label space be $\mathcal{Y} = [k]$, where $d$ is a positive integer denotes the input dimension. $[k] = \{1, 2, \ldots, k\}$ where $k$ denotes the number of classes. Let $\boldsymbol{x} \in \mathcal{X}$ be an instance and $y \in \mathcal{Y}$ be a label, and each example $(\boldsymbol{x}, y) \in \mathcal{X} \times \mathcal{Y}$ is assumed to be drawn from an underlying joint distribution with probability density $p(\boldsymbol{x}, y)$. The goal of multi-class classification is to learn a multi-class classifier $f : \mathcal{X} \mapsto \mathbb{R}^k$ that minimizes the *classification risk* defined as

$$R(f) = \mathbb{E}_{p(\boldsymbol{x}, y)}[\mathcal{L}(f(\boldsymbol{x}), y)], \tag{1}$$

where $\mathbb{E}_{p(\boldsymbol{x}, y)}$ denotes the expectation over $p(\boldsymbol{x}, y)$, and $\mathcal{L} : \mathbb{R}^k \times \mathcal{Y} \mapsto \mathbb{R}_+$ denotes a multi-class classification loss (e.g., the softmax cross entropy loss), which measures the discrepancy between the classifier output $f(\boldsymbol{x})$ and the true label $y$. Given the classifier $f$, the predicted label is given by $\widehat{y} = \mathrm{argmax}_{j \in [k]} f_j(\boldsymbol{x})$, where $f_j(\boldsymbol{x})$ is the $j$-th element of $f(\boldsymbol{x})$.

## 2.2 Learning from Label Proportions

LLP (Yu et al., 2014; Liu et al., 2019) considers a problem where the supervision is given to a set of data and the supervision is the proportion of instances from each class in the set. This supervision provides similar information compared with class priors. However, MCMU and LLP are significantly different from each other, and here we describe the differences between them:

- **From the distribution perspective**: In MCMU, the unlabeled sets are sampled from distributions with different class priors, while the unlabeled sets in LLP are sampled from the same distribution.

- **From the generation perspective**: In MCMU, the generation process of a data point is: class priors $\rightarrow$ ground truth label $y \rightarrow$ data point $\boldsymbol{x}$, while in LLP, the generation process is: a set of data points $X \rightarrow$ a set of ground truth label $y \rightarrow$ the proportion supervision.

- **From the dependence perspective**: In MCMU, given the class priors, the data points in the same set are independent on each other, while in LLP, given the label proportions, the data points in the same set are dependent from each other.

Most previous studies on LLP try to solve this problem by the *empirical proportion risk minimization* (ERPM) (Dery et al., 2017; Tsai & Lin, 2020), which aims to minimize the distance between the approximated label proportion by the classification model and the truth label proportion. There are various approaches to measuring this distance, including squared error or KL-divergence for binary classification (Dery et al., 2017; Tsai & Lin, 2020) and cross-entropy for multi-class classification (Dulac-Arnold et al., 2019; Liu et al., 2019).

## 2.3 Previous Studies on Classification from Unlabeled Datasets

Recent studies (Lu et al., 2019; Luo & Orabona, 2010) showed that it is possible to train a binary classifier from only two unlabeled datasets with different class priors. Hence, many researchers began to investigate classification from unlabeled datasets (Scott & Zhang, 2020; Lu et al., 2021; Tang et al., 2022) and proposed effective solutions. However, compared with our study in this paper, these previous methods normally have the following limitations:

- **Limited to the binary classification**: Most previous methods focus on the binary classification setting (Menon et al., 2015; Lu et al., 2019; Luo & Orabona, 2010; Zhang et al., 2020b; Scott & Zhang, 2020; Lu et al., 2021) and cannot deal with the multi-class classification setting.

- **Limited to the specific number of unlabeled sets**: Some methods are limited to a specific number of unlabeled sets, such as the number of unlabeled sets $m = 2$ (Lu et al., 2019; Luo & Orabona, 2010) or $m = 2k$ ($k \in \mathbb{N}_+$) (Scott & Zhang, 2020).

- **Limited by the negative empirical risk**: Some methods would encounter the negative empirical risk issue (Lu et al., 2019; Tsai & Lin, 2020; Tang et al., 2022), i.e., the risk could empirically become negative during the training process. This poses a serious challenge to the optimization process since it is problematic to minimize an objective that can be unbounded from below.

In our work, we focus on multi-class classification from multiple unlabeled sets and the empirical risk of our proposed methods would never become negative. Table 1 shows the comparisons of our work with previous studies on relevant problem settings.

## 2.4 Multi-Class Classification from Multiple Unlabeled Datasets

Given $m$ ($m \geqslant 2$) sets of unlabeled data $\mathcal{D} := \bigcup_{i=1}^{m} \mathcal{U}_i$ where $\mathcal{U}_i = \{\boldsymbol{x}_{i,r}\}_{r=1}^{n_i}$ is a collection of $n_i$ data points drawn from a mixture of class-conditional densities:

$$\mathcal{U}_i \overset{\text{i.i.d}}{\sim} \bar{p}_i(\boldsymbol{x}) = p(\boldsymbol{x} \mid \bar{y} = i) = \sum_{j=1}^{k} \theta_{ij} \cdot p(\boldsymbol{x} \mid y = j),$$

where $\boldsymbol{\theta} \in \mathbb{R}^{m \times k}$ and $0 \leqslant \theta_{ij} \leqslant 1$ denotes the $j$-th class prior of the $i$-th unlabeled set, and $\bar{y} \in \{1, 2, \ldots, m\}$ denotes the index of the $m$ sets of unlabeled instances.

We assume that these class priors form a full column rank matrix $\boldsymbol{\theta} := (\theta_{ij}) \in [0, 1]^{m \times k}$ with the constraint $\sum_{j=1}^{k} \theta_{ij} = 1$. Let $\boldsymbol{\pi}$ denote the original class priors (the class priors over all $m$ unlabeled sets), i.e., $\pi_j = p(y = j)$ and $\rho_j$ denote the probability of a data point belonging to the $i$-th set, i.e., $\rho_i = p(\bar{y} = i)$. Throughout the paper, we assume that the class priors of each unlabeled set are given, which means $\boldsymbol{\theta}$ is accessible. Then, $\pi_j$ and $\rho_j$ could be estimated by $\sum_{i=1}^{m} n_i \cdot \theta_{ij} / \sum_{i=i}^{m} n_i$ and $n_j / \sum_{i=1}^{m} n_i$ respectively. Although we can only access the unlabeled training sets $\bigcup_{i=1}^{m} \mathcal{U}_i$, our goal in this paper is to learn a multi-class classifier that can make accurate predictions on unseen individual instances.

## 3 The Proposed Consistent Learning Methods

In this section, we present our two statistically consistent methods, i.e., the *classifier-consistent method* (CCM) and the *risk-consistent method* (RCM). We say a method is classifier-consistent if the learned classification model by this method would converge to the optimal classification model obtained by minimizing the classification risk in Eq. (1) and we say a method is risk-consistent if this method holds a risk function that is exactly equivalent to the classification risk in Eq. (1). CCM can be viewed as a multi-class extension of Lu et al. (2021), following which we also consider the MCMU problem as a surrogate set classification problem, i.e., to predict which unlabeled set an instance belongs to. Then we theoretically showed that we can transform the posterior probabilities of ordinary labels to the posterior probabilities of surrogate sets, via a probability transition function. In this way, we can obtain the desired multi-class classifier from the learned surrogate set classifier. However, CCM suffers from both theoretical and practical limitations because it cannot guarantee risk consistency and lacks purified supervision information.

Therefore, we further propose a *risk-consistent method* (RCM). Concretely, we rewrite the risk shown in Eq. (1) into an equivalent form via importance weighting, which can well handle the given unlabeled datasets. RCM dynamically calculates the importance weights during the training process, which enables RCM to differentiate the true label of each training example and progressively purify supervision information during the training process. It is noteworthy that U-PRR (Tang et al., 2022) is also a method that holds risk consistency, which rewrites the classification risk by a pseudo-inverse matrix. However, our proposed RCM is superior to U-PRR according to the following aspects: 1) RCM utilizes both the classifier information and the prior matrix information to dynamically calculate importance weights while U-PRR only statically calculates weights solely based on the prior matrix. 2) U-PRR suffers from the issue of negative empirical risk, while RCM intrinsically avoids this issue by using a non-negative risk estimator. 3) RCM does not involve any hyper-parameters in the training objective while U-PRR needs to tune several hyper-parameters for better training performance. 4) RCM consistently exhibits superior performance across various settings and demonstrates a substantial performance advantage in random matrix settings.

In many popular WSL (Weakly Supervised Learning) problems (such as partial-label learning (Wang et al., 2022), noisy label learning (Han et al., 2018), and multiple-instance learning (Zhou et al., 2009)), we could usually have access to weakly supervised labels. However, in the MCMU problem, neither a ground-truth label nor a weakly supervised label is accessible for each training example, which makes the classification risk in Eq. (1) cannot be easily recovered from the unlabeled sets. To deal with this challenge, Lu et al. (2021) treated the indexes of the $m$ sets as surrogate labels, which would be convenient for deriving a meaningful training objective.

Following this strategy, by treating $\bar{y}$ as a surrogate label, we could transform the $m$ sets unlabeled data $\bigcup_{i=1}^{m} \mathcal{U}_i$ into $\widetilde{\mathcal{D}} = \{(\boldsymbol{x}^{(r)}, \bar{y}^{(r)})\}_{r=1}^{n}$ where $n$ is the total number of data points. Then, we could bridge the two posterior probabilities $p(y \mid \boldsymbol{x})$ and $p(\bar{y} \mid \boldsymbol{x})$ by the following lemma:

**Lemma 3.1.** *Let $\boldsymbol{\eta}(\boldsymbol{x})$ and $\bar{\boldsymbol{\eta}}(\boldsymbol{x})$ be $k$-dimensional and $m$-dimensional probabilistic vectors respectively, where $\eta_j(\boldsymbol{x})$ and $\bar{\eta}_i(\boldsymbol{x})$ denote $p(y = j \mid \boldsymbol{x})$ and $p(\bar{y} = i \mid \boldsymbol{x})$ respectively. Then we have*

$$\bar{\eta}_i(\boldsymbol{x}) = \frac{\rho_i \cdot [\sum_{j=1}^k \theta_{ij} \cdot \frac{\eta_j(\boldsymbol{x})}{\pi_j}]}{\sum_{i=1}^m \rho_i \cdot [\sum_{j=1}^k \theta_{ij} \cdot \frac{\eta_j(\boldsymbol{x})}{\pi_j}]} = \frac{\rho_i \cdot \boldsymbol{\alpha}_i^\top \boldsymbol{\eta}(\boldsymbol{x})}{\boldsymbol{\beta}^\top \boldsymbol{\eta}(\boldsymbol{x})}, \tag{2}$$

*where $\boldsymbol{\alpha}_i = [\alpha_{i1}, \ldots, \alpha_{ik}]$ and $\alpha_{ij} = \frac{\theta_{ij}}{\pi_j}$ and $\boldsymbol{\beta} = \sum_{i=1}^m \rho_i \cdot \boldsymbol{\alpha}_i$.*

The detailed derivation of Lemma 3.1 is provided in Appendix A.

## 3.1 CLASSIFIER-CONSISTENT METHOD VIA PROBABILITY TRANSITION FUNCTION

For the classifier-consistent method, we aim to solve the surrogate set classification problem by standard multi-class classification method. We use $\eta_j(\boldsymbol{x})$ to denote the probability $p(y = j \mid \boldsymbol{x})$, which can be approximated by the softmax output of the classification model. As shown in Eq. (2), $\bar{\eta}_j(\boldsymbol{x})$ could be obtained by substituting $\boldsymbol{\eta}(\boldsymbol{x})$. We define $\boldsymbol{T}(\cdot) : \mathbb{R}^k \to \mathbb{R}^m$ as a transition function to represent this transformation, i.e., $\bar{\boldsymbol{\eta}}(\boldsymbol{x}) = \boldsymbol{T}(\boldsymbol{\eta}(\boldsymbol{x}))$. It is noteworthy that the coefficients in $T_i(\cdot)$ are all constant as shown in Eq. (2) and $\boldsymbol{T}(\cdot)$ is deterministic. Then the learning objective of CCM could be defined as

$$R_{\text{ccm}}(f) = \mathbb{E}_{p(x,\bar{y})}[\mathcal{L}(\bar{\boldsymbol{\eta}}(\boldsymbol{x}), \bar{y})]. \tag{3}$$

It is worth noting that the loss function $\mathcal{L}$ here needs to handle probability vectors (e.g., the cross entropy loss without the softmax processing). In this way, the empirical version of $R_{\text{ccm}}(f)$ can be expressed as

$$\widehat{R}_{\text{ccm}}(f) = \frac{1}{n} \left( \sum_{i=1}^n \mathcal{L}(\bar{\boldsymbol{\eta}}(\boldsymbol{x}^{(i)}), \bar{y}^{(i)}) \right). \tag{4}$$

The pseudo-code of the CCM is provided in Algorithm 1.

Then, in order to prove that this method is classifier-consistent, we introduce the following lemmas:

**Lemma 3.2.** *Let $\theta_{ij} \neq \theta_{is}$, $\forall i \in [1, \ldots, m]$ and $\forall j, s \in [1, \ldots, k]$. Then, the transition function $\boldsymbol{T}(\cdot)$ is an injective function in the domain $[0, 1]$.*

The proof is provided in Appendix B.1. A similar proof can also be found in Lu et al. (2021).

**Lemma 3.3.** *If certain loss functions are used (e.g., the softmax cross entropy loss), by minimizing the expected risk $R(f)$, the optimal mapping $g^\star$ satisfies $g_i^\star(\boldsymbol{x}) = p(y = i | \boldsymbol{x})$.*

The proof is provided in Appendix B.2. The same proof can also be found in (Yu et al., 2018; Feng et al., 2020). Then, we have the following theorem.

**Theorem 3.4.** *When the conditions in Lemma 3.2 and Lemma 3.3 are satisfied, the minimizer $f_{\text{ccm}} = \arg\min_{f \in \mathcal{F}} R_{\text{ccm}}(f)$ is also the true minimizer $f^\star = \arg\min_{f \in \mathcal{F}} R(f)$, i.e., $f_{\text{ccm}} = f^\star$ (classifier consistency).*

The proof is provided in Appendix B.3.

Let $\widehat{f}_{\text{ccm}} = \arg\min_{f \in \mathcal{F}} \widehat{R}_{\text{ccm}}(f)$ be the empirical risk minimizer, and $f^\star = \arg\min_{f \in \mathcal{F}} R(f)$ be the true risk minimizer. Besides, we define the function space $\mathcal{H}_y$ for the label $y \in \mathcal{Y}$ as $\{h : \boldsymbol{x} \to f_y(\boldsymbol{x}) \mid f \in \mathcal{F}\}$. Let $\mathfrak{R}_n(\mathcal{H}_y)$ be the expected Rademacher complexity (Bartlett & Mendelson, 2002) of $\mathcal{H}_y$ with sample size $n$, then we have the following theorem.

**Theorem 3.5.** *Let the loss function $\mathcal{L}(\boldsymbol{T}(\boldsymbol{\eta}(\boldsymbol{x})), \bar{y})$ be $L'$-Lipschitz with respect to $f(\boldsymbol{x})$ ($0 < L' < \infty$) and bounded by $C'_\ell$, i.e., $\sup_{\boldsymbol{x} \in \mathcal{X}, f \in \mathcal{F}, \bar{y} \in [m]} \mathcal{L}(\boldsymbol{T}(\boldsymbol{\eta}(\boldsymbol{x})), \bar{y}) \leqslant C'_\ell$. Then for any $\delta > 0$, with probability at least $1 - \delta$,*

$$R_{\text{ccm}}(\widehat{f}_{\text{ccm}}) - R_{\text{ccm}}(f^\star) \leqslant 4\sqrt{2}L' \sum_{y=1}^k \mathfrak{R}_n(\mathcal{H}_y) + 2C'_\ell \sqrt{\frac{\log\frac{2}{\delta}}{2n}}.$$

The proof is provided in Appendix B.4. Generally, $\mathfrak{R}_n(\mathcal{H}_y)$ can be upper bounded by $\mathcal{C}_\mathcal{H}/\sqrt{n}$ for a positive constant $\mathcal{C}_\mathcal{H}$ (Lu et al., 2020; Golowich et al., 2017). Theorem 3.5 demonstrates that the empirical minimizer $\widehat{f}_{\text{ccm}}$ learned by CCM would convergent to the minimizer $f^\star$ learned from clean data as the training sample size approaches infinity.

## 3.2 RISK-CONSISTENT METHOD VIA IMPORTANCE WEIGHTING

For the risk-consistent method, our strategy is to solve the MCMU problem by risk rewriting (Gretton et al., 2009), i.e., transforming the classification risk into an equivalent form that can be accessible from multiple unlabeled sets with class priors.

**Theorem 3.6.** *The classification risk $R(f)$ in Eq. (1) can be equivalently expressed as follows:*

$$R_{\mathrm{rcm}}(f) = \mathbb{E}_{p(\boldsymbol{x},\bar{y})}\big[\textstyle\sum_{j=1}^{k} p(y=j \mid \bar{y}, \boldsymbol{x})\mathcal{L}(f(\boldsymbol{x}), j)\big]. \tag{5}$$

The proof of Theorem 3.6 is provided in Appendix C.1. It is worth noting that the loss function $\mathcal{L}$ here needs to handle logit vectors before the softmax processing (e.g., the softmax cross entropy loss). It can be observed that the $R_{\mathrm{rcm}}$ works as an importance-weighting loss due to the probability function $p(y = j \mid \bar{y}, \boldsymbol{x})$, and $p(y = j \mid \bar{y}, \boldsymbol{x})$ indicates the probability of $\boldsymbol{x}$ in $\bar{y}$-th unlabeled set belonging to class $j$.

Here, $p(y = j \mid \bar{y}, \boldsymbol{x})$ can be calculated by

$$p(y = j \mid \bar{y}, \boldsymbol{x}) = \frac{p(\boldsymbol{x} \mid y=j, \bar{y})p(y=j, \bar{y})}{p(\bar{y}, \boldsymbol{x})} = \frac{p(\boldsymbol{x} \mid y=j)p(y=j \mid \bar{y})p(\bar{y})}{p(\bar{y}, \boldsymbol{x})}$$

$$= \frac{\frac{p(y=j|\boldsymbol{x})}{p(y=j)}p(y=j \mid \bar{y})p(\bar{y})}{p(\bar{y} \mid \boldsymbol{x})} = \frac{\frac{\eta_j(\boldsymbol{x})}{\pi_j}\theta_{\bar{y}j}\rho_{\bar{y}}}{\bar{\eta}_{\bar{y}}(\boldsymbol{x})} = \frac{\eta_j(\boldsymbol{x})\alpha_{\bar{y}j}\boldsymbol{\beta}^{\top}\boldsymbol{\eta}(\boldsymbol{x})}{\boldsymbol{\alpha}_{\bar{y}}^{\top}\boldsymbol{\eta}(\boldsymbol{x})}. \tag{6}$$

The above derivation uses the fact that $p(\boldsymbol{x} \mid y = j) = p(\boldsymbol{x} \mid y = j, \bar{y})$, since we could obtain the underlying class-conditional distribution independent from the surrogate label, given the true label. After obtaining the formulation of $R_{\mathrm{rcm}}$, its empirical version $\widehat{R}_{\mathrm{rcm}}$ can be expressed as

$$\widehat{R}_{\mathrm{rcm}} = \textstyle\sum_{v=1}^{n} \Big( \textstyle\sum_{j=1}^{k} p(y=j \mid \bar{y}^{(v)}, \boldsymbol{x}^{(v)})\mathcal{L}(f(\boldsymbol{x}^{(v)}), j) \Big). \tag{7}$$

Now, the problem becomes how to accurately estimate $p(y = j \mid \bar{y}, \boldsymbol{x})$, since cannot not be directly accessible from the unlabeled datasets. We apply the softmax function on the outputs of the classification model $f(\boldsymbol{x})$ to approximate $\boldsymbol{\eta}(\boldsymbol{x})$, concretely, we use $\exp(f_j(\boldsymbol{x}))/\sum_{i=1}^{k} \exp(f_i(\boldsymbol{x}))$ to approximate $\eta_j(\boldsymbol{x})$. Then, by substituting the approximated $\boldsymbol{\eta}(\boldsymbol{x})$ into Eq. (6), we could obtain the estimated $p(y = j \mid \bar{y}, \boldsymbol{x})$ due to Eq. (6). The pseudo-code of RCM is shown in Algorithm 2.

From the theoretical perspective, RCM tries to recover the ordinary classification risk defined in Eq. (1) by using the estimated probability function $p(y = j \mid \bar{y}, \boldsymbol{x})$. Hence we do not impose any restrictions on the loss function $\mathcal{L}$ and the classification model $f$, and we could utilize classification models in RCM. From the practical perspective, RCM works in an importance-weighting manner. RCM synthesizes the surrogate label information and the classification model information to approximate the distribution of the ground truth label $y$. Specifically, when the estimated $p(y = j \mid \bar{y}, \boldsymbol{x})$ is small, the label $j$ is unlikely to be the ground truth label of the instance $\boldsymbol{x}$, then the weight $p(y = j \mid \bar{y}, \boldsymbol{x})$ on the loss $\mathcal{L}(f(\boldsymbol{x}), j)$ is small. In this way, the weights help the classification model identify the ground truth label and purify supervision information while the classification model also helps the method to obtain better estimated weights. These two procedures work alternately and reciprocate each other. We further provide a more detailed theoretical analysis of the influence between weights and classification models based on the EM algorithm. The analysis is provided in Appendix C.2.

Furthermore, we derive a generalization error bound to theoretically analyze our RCM. Let $\widehat{f}_{\mathrm{rcm}} = \arg\min_{f \in \mathcal{F}} \widehat{R}_{\mathrm{rcm}}(f)$ be the empirical risk minimizer. Besides, we define the function space $\mathcal{H}_y$ for the label $y \in \mathcal{Y}$ as $\{h : \boldsymbol{x} \to f_y(\boldsymbol{x}) \mid f \in \mathcal{F}\}$. Let $\mathfrak{R}_n(\mathcal{H}_y)$ be the expected Rademacher complexity (Bartlett & Mendelson, 2002) of $\mathcal{H}_y$ with sample size $n$, then we have the following theorem.

**Theorem 3.7.** *Let the used multi-class classification loss function $\mathcal{L}(f(\boldsymbol{x}), y)$ be $L$-Lipschitz with respect to $f(\boldsymbol{x})$ $(0 < L < \infty)$ and bounded by $C_\ell$, i.e., $\sup_{\boldsymbol{x} \in \mathcal{X}, f \in \mathcal{F}, y \in \mathcal{Y}} \mathcal{L}(f(\boldsymbol{x}), y) \leqslant C_\ell$. Then for any $\delta > 0$, with probability at least $1 - \delta$,*

$$\mathbb{E}\big[\widehat{R}_{\mathrm{rcm}}(\widehat{f}_{\mathrm{rcm}})\big] - \widehat{R}_{\mathrm{rcm}}(\widehat{f}_{\mathrm{rcm}}) \leqslant 2\sqrt{2}L \sum_{y=1}^{k} \mathfrak{R}_n(\mathcal{H}_y) + C_\ell \sqrt{\frac{\log\frac{1}{\delta}}{2n}}.$$

---

**Algorithm 1** CCM Algorithm

---

**Input:** Model $f$, class prior matrix $\boldsymbol{\theta}$, original class prior $\boldsymbol{\pi}$, surrogate class prior $\boldsymbol{\rho}$, epoch $E_{\max}$, iteration $I_{\max}$, unlabeled training set $\widetilde{\mathcal{D}} = \{(\boldsymbol{x}^{(i)}, \bar{y}^{(i)})\}_{i=1}^n$;

1: **for** $e = 1, 2, \ldots, E_{\max}$ **do**
2:      **Shuffle** the unlabeled training set $\widetilde{\mathcal{D}} = \{(\boldsymbol{x}^{(i)}, \bar{y}^{(i)})\}_{i=1}^n$;
3:      **for** $j = 1, \ldots, I_{\max}$ **do**
4:          **Fetch** mini-batch $\widetilde{\mathcal{D}}_j$ from $\widetilde{\mathcal{D}}$;
5:          **Calculate** estimated $\bar{\boldsymbol{\eta}}(\boldsymbol{x}^{(i)})$ by substituting estimated $\boldsymbol{\eta}(\boldsymbol{x}^{(i)})$ into Eq. (2)
6:          **Update** model $f$ by minimizing the empirical risk estimator $\widehat{R}_{\text{ccm}}$ in Eq. (4);
7:      **end for**
8: **end for**
**Output:** $f$.

---

**Algorithm 2** RCM Algorithm

---

**Input:** Model $f$, class prior matrix $\boldsymbol{\theta}$, original class prior $\boldsymbol{\pi}$, surrogate class prior $\boldsymbol{\rho}$, epoch $E_{\max}$, warmup epoch $E_{\text{warm}}$, iteration $I_{\max}$, unlabeled training set $\widetilde{\mathcal{D}} = \{(\boldsymbol{x}^{(i)}, \bar{y}^{(i)})\}_{i=1}^n$.

1: **Warmup** $f$ by CCM for $E_{\text{warm}}$ epochs;
2: **Initialize** estimated $p(y|\boldsymbol{x}^{(i)}, \bar{y}^{(i)})$ by Eq. (6);
3: **for** $e = 1, 2, \ldots, E_{\max}$ **do**
4:      **Shuffle** $\widetilde{\mathcal{D}} = \{(\boldsymbol{x}^{(i)}, \bar{y}^{(i)})\}_{i=1}^n$;
5:      **for** $j = 1, \ldots, I_{\max}$ **do**
6:          **Fetch** mini-batch $\widetilde{\mathcal{D}}_j$ from $\widetilde{\mathcal{D}}$;
7:          **Update** model $f$ by $\widehat{R}_{\text{rcm}}$ in Eq. (7);
8:          **Update** estimated $p(y|\boldsymbol{x}^{(i)}, \bar{y})$ by Eq. (6);
9:      **end for**
10: **end for**      **Output:** $f$.

---

The proof of Theorem 3.7 is provided in Appendix C.3. Theorem 3.7 demonstrates that the training error $\widehat{R}_{\text{rcm}}(\widehat{f}_{\text{rcm}})$ would convergent to the generalization error $\mathbb{E}[\widehat{R}_{\text{rcm}}(\widehat{f}_{\text{rcm}})]$ as the the training sample size approaches infinity.

**Comparisions Between RCM and CCM.** From a theoretical perspective, RCM could approximate or simulate the distribution of real clean data by utilizing the data distribution from the unlabeled set (by the importance-weighting schema). This means that RCM attempts to infer latent distribution patterns similar to those of real clean data from the unlabeled data. In contrast, CCM aims to better fit the distribution of the unlabeled set by maximizing a log-likelihood object. With a sufficient number of samples, RCM is more accurate in predicting the labels of unseen samples because it considers the restoration of the distribution of real clean data when modeling unlabeled data. This enables RCM to exhibit better generalization performance when facing unknown data, making more precise predictions for unseen samples. More detailed descriptions and explanations of RCM and CCM are provided in Appendix D.

## 4 EXPERIMENTS

### 4.1 EXPERIMENTAL SETUP

**Datasets.** We use 5 popular benchmark datasets including MNIST (LeCun et al., 1998), Kuzushiji-MNIST (Clanuwat et al., 2018), Fasion-MNIST (Xiao et al., 2017), CIFAR10 (Krizhevsky et al., 2009) and SVHN (Netzer et al., 2011). In the experiments, the number of data points in all sets is the same, and the total number of training data points is fixed, i.e., $n_1 = n_2 = \cdots n_m$ and $\sum_{i=1}^m n_i$ is equal to the size of the training dataset. The $n_i$ data points contained in $i$-th unlabeled set were randomly sampled according to $\theta_{i,1}, \theta_{i,2}, \cdots, \theta_{i,m}$ without replacement. Since our methods is flexible on classification models, we apply 5-layer MLP on MNIST, Kuzushiji-MNIST and Fashion-MNIST and ResNet (He et al., 2016) is used on CIFAR10 and SVHN. For fair comparison, we apply the same classification model on the same dataset for each method. More training details are reported in Appendix E.1.

**Class prior matrix.** To better analyze the performance of our proposed methods, we construct different class prior matrices $\boldsymbol{\theta}$ in experiments. Previous studies on MCMU conduct experiments on the diagonal-dominated matrix or the matrix conducted by two diagonal-dominated matrices. For the m sets generated according to such a matrix, there exists at least one set $\mathcal{U}_i$ that the class $j$ accounts for the largest portion in $\mathcal{U}_i$. However, this setting is hard to apply in real-world scenarios. To better simulate real-world situations, we conduct experiments on both the diagonal-dominated matrix and the non-diagonal-dominated random matrix.

Table 2: Classification accuracy (mean±std) for each methods on $m = 10$. Symmetric, Asymmetric, and Random in the table refer to Symmetric diagonal-dominated matrix, Asymmetric diagonal-dominated matrix, and Random matrix respectively. The best and comparable methods based on the paired $t$-test at the significance level 5% are highlighted in boldface.

| Matrix | Datasets | MNIST | Fashion | Kuzushiji | CIFAR-10 | SVHN |
|--------|----------|-------|---------|-----------|----------|------|
| | Supervised | $98.15 \pm 0.05\%$ | $88.39 \pm 0.18\%$ | $90.98 \pm 0.18\%$ | $70.21 \pm 0.43\%$ | $92.68 \pm 0.29\%$ |
| Symmetric | Unbiased | $25.34 \pm 22.65\%$ | $29.40 \pm 10.39\%$ | $15.81 \pm 12.21\%$ | $13.32 \pm 11.71\%$ | $11.35 \pm 16.07\%$ |
| | U-Stop | $81.44 \pm 6.91\%$ | $74.40 \pm 5.40\%$ | $54.16 \pm 8.27\%$ | $34.72 \pm 5.64\%$ | $25.01 \pm 11.06\%$ |
| | U-Correct | $86.20 \pm 6.21\%$ | $76.82 \pm 3.98\%$ | $66.89 \pm 6.25\%$ | $39.14 \pm 3.32\%$ | $68.09 \pm 9.88\%$ |
| | U-Flood | $83.93 \pm 4.93\%$ | $77.33 \pm 3.36\%$ | $66.74 \pm 5.23\%$ | $36.51 \pm 3.00\%$ | $69.46 \pm 7.18\%$ |
| | Prop | $85.99 \pm 5.84\%$ | $75.16 \pm 5.02\%$ | $73.32 \pm 4.97\%$ | $51.21 \pm 4.12\%$ | $74.12 \pm 7.43\%$ |
| | U-PRR | $80.37 \pm 6.98\%$ | $72.30 \pm 6.97\%$ | $67.79 \pm 7.54\%$ | $43.39 \pm 5.42\%$ | $69.08 \pm 8.51\%$ |
| | CCM | $94.03 \pm 1.47\%$ | $\mathbf{83.37 \pm 1.88}\%$ | $\mathbf{79.06 \pm 3.73}\%$ | $60.05 \pm 5.01\%$ | $80.80 \pm 5.81\%$ |
| | RCM | $\mathbf{94.28 \pm 2.26}\%$ | $82.81 \pm 3.13\%$ | $78.91 \pm 3.87\%$ | $\mathbf{65.68 \pm 4.88}\%$ | $\mathbf{84.22 \pm 6.96}\%$ |
| | Supervised | $98.24 \pm 0.15\%$ | $88.68 \pm 0.12\%$ | $91.03 \pm 0.20\%$ | $71.91 \pm 0.85\%$ | $92.89 \pm 0.25\%$ |
| Asymmetric | Unbiased | $82.82 \pm 2.01\%$ | $70.53 \pm 1.45\%$ | $57.70 \pm 4.52\%$ | $48.65 \pm 1.04\%$ | $69.57 \pm 2.41\%$ |
| | U-Stop | $89.08 \pm 2.18\%$ | $79.51 \pm 0.59\%$ | $68.38 \pm 4.61\%$ | $48.38 \pm 0.43\%$ | $65.99 \pm 3.42\%$ |
| | U-Correct | $94.56 \pm 0.49\%$ | $83.43 \pm 0.27\%$ | $79.41 \pm 0.74\%$ | $49.85 \pm 1.87\%$ | $85.77 \pm 0.62\%$ |
| | U-Flood | $92.85 \pm 0.26\%$ | $82.44 \pm 0.59\%$ | $76.56 \pm 1.24\%$ | $50.60 \pm 0.95\%$ | $83.14 \pm 0.58\%$ |
| | Prop | $96.13 \pm 0.22\%$ | $\mathbf{86.46 \pm 0.37}\%$ | $84.48 \pm 0.95\%$ | $61.60 \pm 1.33\%$ | $85.14 \pm 0.53\%$ |
| | U-PRR | $94.44 \pm 0.36\%$ | $85.15 \pm 0.28\%$ | $81.94 \pm 0.47\%$ | $56.34 \pm 1.16\%$ | $84.91 \pm 0.74\%$ |
| | CCM | $96.08 \pm 0.22\%$ | $85.24 \pm 0.43\%$ | $85.01 \pm 0.28\%$ | $67.73 \pm 1.84\%$ | $88.25 \pm 0.77\%$ |
| | RCM | $\mathbf{96.75 \pm 0.11}\%$ | $85.87 \pm 0.15\%$ | $\mathbf{85.65 \pm 0.31}\%$ | $\mathbf{74.55 \pm 0.27}\%$ | $\mathbf{92.03 \pm 0.40}\%$ |
| | Supervised | $97.97 \pm 0.13\%$ | $88.10 \pm 0.10\%$ | $90.60 \pm 0.13\%$ | $70.86 \pm 0.63\%$ | $92.43 \pm 0.22\%$ |
| Random | Unbiased | $14.25 \pm 4.55\%$ | $13.04 \pm 2.16\%$ | $11.9 \pm 0.82\%$ | $12.56 \pm 2.01\%$ | $10.11 \pm 1.70\%$ |
| | U-Stop | $22.67 \pm 4.81\%$ | $46.70 \pm 23.55\%$ | $24.92 \pm 14.76\%$ | $18.63 \pm 2.34\%$ | $8.49 \pm 1.20\%$ |
| | U-Correct | $25.23 \pm 9.32\%$ | $28.45 \pm 10.40\%$ | $17.48 \pm 4.44\%$ | $14.21 \pm 3.55\%$ | $11.13 \pm 2.92\%$ |
| | U-Flood | $78.95 \pm 16.06\%$ | $71.79 \pm 8.38\%$ | $53.68 \pm 16.12\%$ | $25.88 \pm 3.63\%$ | $13.12 \pm 4.50\%$ |
| | Prop | $89.09 \pm 1.76\%$ | $79.26 \pm 2.25\%$ | $67.12 \pm 2.58\%$ | $37.38 \pm 2.75\%$ | $59.56 \pm 3.80\%$ |
| | U-PRR | $26.56 \pm 3.79\%$ | $14.25 \pm 3.53\%$ | $21.79 \pm 2.45\%$ | $17.71 \pm 2.37\%$ | $10.05 \pm 2.31\%$ |
| | CCM | $91.92 \pm 1.19\%$ | $78.07 \pm 6.38\%$ | $72.62 \pm 2.33\%$ | $42.31 \pm 2.99\%$ | $72.73 \pm 4.70\%$ |
| | RCM | $\mathbf{95.04 \pm 0.58}\%$ | $\mathbf{81.11 \pm 2.13}\%$ | $\mathbf{78.60 \pm 2.61}\%$ | $\mathbf{50.51 \pm 4.59}\%$ | $\mathbf{78.70 \pm 7.88}\%$ |

- **Symmetric diagonal-dominated square matrix:** In this matrix, the largest element in each row is the element on the diagonal, and the rest elements in the same row are completely identical. Concretely, the elements $\theta_{ij}$ are uniformly sampled from $[1/k, 1]$ when $i = j$, and set $\theta_{ij} = (1 - \theta_{i,i})/(k-1)$ when $i \neq j$.

- **Asymmetric diagonal-dominated square matrix:** In this matrix, the elements on the diagonals of $\boldsymbol{\theta}$ are larger than the elements in the same row, i.e., $\theta_{i,i} > \theta_{ij}, 1 \leqslant i, j \leqslant m, i \neq j$ and the non-diagonal elements in the same row may be not identical. Concretely, the class priors $\theta_{ij}$ are uniformly generated from $[1, 1/m]$ when $i \neq j$, and $\theta_{i,i} = 1 - \sum_{j \neq i} \theta_{ij}$.

- **Random matrix:** This matrix is a totally random matrix. Firstly, we uniformly generate each element in the matrix from $[0, 1]$, then we assign each element $\theta_{ij}$ the value $\theta_{ij}/\sum_{v=1}^{k} \theta_{iv}$ to ensure the summation of each row is equal to 1.

We conduct experiments not only on the square matrix but also on the non-square matrix when $m > k$. Specifically, we set $m = 2k$ and the non-square class-prior matrix is constructed by concatenating two squared matrices described above.

**Compared methods.** We compared our proposed methods with the following methods including Unbiased (Tang et al., 2022), U-Stop (Tang et al., 2022), U-Correct (Tang et al., 2022), U-Flood (Tang et al., 2022), Prop (Yu et al., 2014), U-PRR (Tang et al., 2022). More detailed descriptions of the compared methods are shown in Appendix E.2.

## 4.2 EXPERIMENTAL RESULTS

The experimental results on $m = 10$ and $m = 20$ based on the 3 class prior matrices are shown in Table. 2 and Table. 3 respectively. To verify the robustness of the methods against the noisy prior matrix $\theta$, variant set numbers $m$ and variant set sizes, we also provide additional experimental results in Appendix F. From Table 2 and Table 3, we could observe that our proposed RCM achieves the best performance and outperforms the other methods in most cases. In addition, RCM and

Table 3: Classification accuracy (mean±std) for each methods on $m = 20$. Symmetric, Asymmetric, and Random in the table refer to Symmetric diagonal-dominated matrix, Asymmetric diagonal-dominated matrix, and Random matrix respectively. The best and comparable methods based on the paired $t$-test at the significance level 5% are highlighted in boldface.

| Matrix | Datasets | MNIST | Fashion | Kuzushiji | CIFAR-10 | SVHN |
|---|---|---|---|---|---|---|
| | Supervised | $98.20 \pm 0.22\%$ | $88.29 \pm 0.16\%$ | $90.64 \pm 0.39\%$ | $70.89 \pm 1.17\%$ | $92.45 \pm 0.25\%$ |
| Symmetric | Unbiased | $57.02 \pm 13.03\%$ | $51.02 \pm 9.17\%$ | $36.42 \pm 6.45\%$ | $30.14 \pm 11.81\%$ | $40.87 \pm 17.05\%$ |
| | U-Stop | $88.48 \pm 1.66\%$ | $80.03 \pm 1.38\%$ | $65.75 \pm 4.41\%$ | $42.82 \pm 2.63\%$ | $44.58 \pm 11.35\%$ |
| | U-Correct | $92.41 \pm 1.41\%$ | $82.06 \pm 1.51\%$ | $74.10 \pm 3.85\%$ | $46.33 \pm 3.15\%$ | $78.43 \pm 5.38\%$ |
| | U-Flood | $90.07 \pm 2.59\%$ | $81.16 \pm 1.44\%$ | $71.80 \pm 3.45\%$ | $44.66 \pm 2.93\%$ | $78.13 \pm 3.29\%$ |
| | Prop | $88.36 \pm 4.70\%$ | $79.71 \pm 3.59\%$ | $74.12 \pm 5.16\%$ | $54.18 \pm 4.54\%$ | $84.66 \pm 2.09\%$ |
| | U-PRR | $85.75 \pm 4.74\%$ | $78.23 \pm 3.79\%$ | $70.08 \pm 6.05\%$ | $47.14 \pm 5.41\%$ | $74.96 \pm 5.34\%$ |
| | CCM | $94.66 \pm 0.90\%$ | $\mathbf{84.07 \pm 1.19}\%$ | $80.51 \pm 2.37\%$ | $59.71 \pm 2.33\%$ | $81.95 \pm 5.14\%$ |
| | RCM | $\mathbf{95.47 \pm 0.72}\%$ | $\mathbf{84.54 \pm 0.70}\%$ | $\mathbf{81.56 \pm 2.17}\%$ | $\mathbf{66.99 \pm 3.28}\%$ | $\mathbf{90.24 \pm 0.93}\%$ |
| | Supervised | $98.09 \pm 0.06\%$ | $88.36 \pm 0.11\%$ | $90.99 \pm 0.32\%$ | $71.34 \pm 0.98\%$ | $92.45 \pm 0.63\%$ |
| Asymmetric | Unbiased | $85.21 \pm 5.27\%$ | $72.69 \pm 2.13\%$ | $62.25 \pm 4.26\%$ | $48.69 \pm 1.01\%$ | $69.23 \pm 2.64\%$ |
| | U-Stop | $90.92 \pm 0.11\%$ | $81.47 \pm 0.90\%$ | $69.69 \pm 0.56\%$ | $48.49 \pm 0.92\%$ | $61.75 \pm 2.30\%$ |
| | U-Correct | $94.30 \pm 0.22\%$ | $83.59 \pm 0.73\%$ | $78.74 \pm 1.01\%$ | $49.67 \pm 1.57\%$ | $84.89 \pm 0.89\%$ |
| | U-Flood | $93.43 \pm 0.49\%$ | $82.68 \pm 0.50\%$ | $76.51 \pm 0.95\%$ | $51.72 \pm 1.44\%$ | $82.24 \pm 1.25\%$ |
| | Prop | $95.84 \pm 0.41\%$ | $\mathbf{85.68 \pm 0.21}\%$ | $82.54 \pm 0.64\%$ | $60.22 \pm 1.17\%$ | $88.73 \pm 0.67\%$ |
| | U-PRR | $92.72 \pm 0.86\%$ | $84.09 \pm 0.80\%$ | $76.98 \pm 1.36\%$ | $51.52 \pm 2.50\%$ | $75.85 \pm 3.06\%$ |
| | CCM | $95.92 \pm 0.35\%$ | $85.18 \pm 0.67\%$ | $83.28 \pm 0.41\%$ | $65.18 \pm 3.47\%$ | $86.99 \pm 1.33\%$ |
| | RCM | $\mathbf{96.33 \pm 0.14}\%$ | $85.37 \pm 0.28\%$ | $\mathbf{84.10 \pm 0.55}\%$ | $\mathbf{69.72 \pm 0.55}\%$ | $\mathbf{91.54 \pm 0.19}\%$ |
| | Supervised | $98.11 \pm 0.06\%$ | $88.31 \pm 0.17\%$ | $91.00 \pm 0.20\%$ | $70.73 \pm 1.16\%$ | $92.40 \pm 0.46\%$ |
| Random | Unbiased | $26.46 \pm 3.68\%$ | $31.36 \pm 2.06\%$ | $17.36 \pm 1.95\%$ | $15.46 \pm 4.45\%$ | $16.46 \pm 5.51\%$ |
| | U-Stop | $55.70 \pm 12.81\%$ | $74.08 \pm 2.71\%$ | $31.20 \pm 19.28\%$ | $31.09 \pm 2.64\%$ | $51.59 \pm 8.16\%$ |
| | U-Correct | $54.34 \pm 4.19\%$ | $60.76 \pm 2.45\%$ | $32.68 \pm 2.67\%$ | $25.29 \pm 2.65\%$ | $17.51 \pm 3.13\%$ |
| | U-Flood | $89.04 \pm 0.73\%$ | $78.85 \pm 1.11\%$ | $67.43 \pm 0.37\%$ | $31.32 \pm 2.28\%$ | $68.70 \pm 3.11\%$ |
| | Prop | $86.73 \pm 1.12\%$ | $79.44 \pm 2.11\%$ | $60.49 \pm 1.50\%$ | $33.26 \pm 2.50\%$ | $53.15 \pm 4.31\%$ |
| | U-PRR | $26.82 \pm 1.74\%$ | $27.45 \pm 1.88\%$ | $16.11 \pm 2.84\%$ | $23.32 \pm 1.36\%$ | $11.01 \pm 2.40\%$ |
| | CCM | $92.21 \pm 0.25\%$ | $81.46 \pm 0.54\%$ | $74.08 \pm 1.07\%$ | $46.23 \pm 1.79\%$ | $66.16 \pm 26.68\%$ |
| | RCM | $\mathbf{95.37 \pm 0.18}\%$ | $\mathbf{83.48 \pm 0.23}\%$ | $\mathbf{80.68 \pm 0.81}\%$ | $\mathbf{55.71 \pm 5.93}\%$ | $\mathbf{88.99 \pm 1.01}\%$ |

CCM outperform the compared methods with a large gap in the random matrix, which verifies the effectiveness of our proposed method in a complex scenario.

Compared to the performance between RCM and CCM, RCM demonstrated superior performance in most settings. When utilizing the same prior matrix, RCM exhibits a larger performance gap over CCM in CIFAR-10 and SVHN, indicating that when applying deeper models, RCM can achieve even more remarkable results. When trained on the same dataset, compared to two diagonal-dominated matrices, RCM demonstrates a larger performance gap over CCM on the Random matrix. This highlights the ability of RCM to handle complex prior matrices more effectively.

**On the variation of set size.** We also conduct experiments to verify the performance of our proposed method on different sizes. Concretely, we randomly select $\epsilon \cdot n_i$ data points from the $i$-th set. The experimental results are provided in Fig. 1 in Appendix F.2. As shown in Fig. 1, RCM and CCM achieve better performance with larger $\epsilon$. Moreover, CCM achieves better performance than RCM when $\epsilon$ is small and RCM outperforms CCM when $\epsilon$ increases, which demonstrates that CCM is more robust when few data points are provided.

## 5 CONCLUSION

In this paper, we studied an interesting problem called *multi-class classification from multiple unlabeled datasets* (MCMU), where the only supervision is the class priors for unlabeled sets. To solve this problem, we first proposed a *classifier-consistent method* based on a probability transition function. Nevertheless, CCM cannot guarantee risk consistency and neglects to differentiate the true labels of training examples. Hence, we further proposed a *risk-consistent method* (RCM) based on importance weighting. RCM can progressively purify supervision information during training. In addition, we provided comprehensive theoretical analyses of the two methods to show the theoretical guarantees. We conducted comprehensive experiments on various settings, and experimental results demonstrated the effectiveness of our proposed methods.

ACKNOWLEDGEMENTS

Lei Feng is supported by the Chongqing Overseas Chinese Entrepreneurship and Innovation Support Program. Bo An is supported by the National Research Foundation Singapore and DSO National Laboratories under the AI Singapore Programme (AISGAward No: AISG2-GC-2023-009). Hongxin Wei is supported by Shenzhen Fundamental Research Program JCYJ20230807091809020.

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

# A  PROOF OF LEMMA 3.1

$\forall i \in \{1, \dots, m\}$ we have

$$\bar{\eta}_i(\boldsymbol{x}) = \frac{p(\boldsymbol{x}, \bar{y} = i)}{\bar{p}(\boldsymbol{x})} = \frac{p(\boldsymbol{x} \mid \bar{y} = i) \cdot p(\bar{y} = i)}{\sum_{i=1}^{m} p(\bar{y} = i) \cdot \bar{p}_i(\boldsymbol{x})} = \frac{\rho_i \cdot [\sum_{j=1}^{k} \theta_{ij} \cdot p(\boldsymbol{x} \mid y = j)]}{\sum_{i=1}^{m} \rho_i \cdot [\sum_{j=1}^{k} \theta_{ij} \cdot p(\boldsymbol{x} \mid y = j)]},$$

where $\bar{p}_i(\boldsymbol{x})$ is used to denotes the mixture probability densities function of set $i$.

And by Bayes' rule we have

$$p(\boldsymbol{x} \mid y = j) = \frac{p(y = j \mid \boldsymbol{x}) \cdot p(\boldsymbol{x})}{p(y = j)} = \frac{\eta_j(\boldsymbol{x}) \cdot p(\boldsymbol{x})}{\pi_j}.$$

Then we can obtain

$$\bar{\eta}_i(\boldsymbol{x}) = \frac{\rho_i \cdot [\sum_{j=1}^{k} \theta_{ij} \cdot \frac{\eta_j(\boldsymbol{x})}{\pi_j}]}{\sum_{i=1}^{m} \rho_i \cdot [\sum_{j=1}^{k} \theta_{ij} \cdot \frac{\eta_j(\boldsymbol{x})}{\pi_j}]} = \frac{\rho_i \cdot \boldsymbol{\alpha}_i^\top \boldsymbol{\eta}(\boldsymbol{x})}{\boldsymbol{\beta}^\top \boldsymbol{\eta}(\boldsymbol{x})}, \tag{8}$$

where $\boldsymbol{\alpha}_i = [\alpha_{i1}, \dots, \alpha_{ik}]$ and $\alpha_{ij} = \frac{\theta_{ij}}{\pi_j}$ and $\boldsymbol{\beta} = \sum_{i=1}^{m} \rho_i \cdot \boldsymbol{\alpha}_i$. $\qquad\square$

# B  PROOFS OF CLASSIFIER-CONSISTENCY METHOD

## B.1  PROOF OF LEMMA 3.2

We proceed the proof by firstly showing that the denominator of each function $T_i(\boldsymbol{t})$, $i \in [1, \dots, m]$, is strictly greater than zero for all $\boldsymbol{t} \in [0, 1]^k$ and then showing that $T_i(\boldsymbol{t}_1) = T_i(\boldsymbol{t}_2)$ if and only if $\boldsymbol{t}_1 = \boldsymbol{t}_2$. For all $i \in [1, \dots, m]$ and $j \in [1, \dots, k]$, $\rho_i > 0$, $\alpha_{ij} > 0$ and $\beta_j > 0$. Then we have the denominator of each function $T_i(\boldsymbol{t})$, $i \in [1, \dots, m]$, is strictly greater than zero for all $\boldsymbol{t} \in [0, 1]^k$. Next, assume that there exist $\boldsymbol{t}_1, \boldsymbol{t}_2 \in [0, 1]^k$ such that $\boldsymbol{t}_1 \neq \boldsymbol{t}_2$ but $\boldsymbol{T}(\boldsymbol{t}_1) = \boldsymbol{T}(\boldsymbol{t}_2)$, which indicates that $T_i(\boldsymbol{t}_1) = T_i(\boldsymbol{t}_2), \forall i \in [1, \dots, m]$. For all $i$, we have

$$\begin{aligned} T_i(\boldsymbol{t}_1) - T_i(\boldsymbol{t}_2) &= \frac{\rho_i \cdot \boldsymbol{\alpha}_i^\top \boldsymbol{t}_1}{\boldsymbol{\beta}^\top \boldsymbol{t}_1} - \frac{\rho_i \cdot \boldsymbol{\alpha}_i^\top \boldsymbol{t}_2}{\boldsymbol{\beta}^\top \boldsymbol{t}_2} \\ &= \rho_i \left( \frac{\boldsymbol{\alpha}_i^\top \boldsymbol{t}_1}{\boldsymbol{\beta}^\top \boldsymbol{t}_1} - \frac{\boldsymbol{\alpha}_i^\top \boldsymbol{t}_2}{\boldsymbol{\beta}^\top \boldsymbol{t}_2} \right) \\ &= 0. \end{aligned}$$

Since $\theta_{ij} \neq \theta_{is}$, $\forall i \in [1, \dots, m]$ and $\forall j, s \in [1, \dots, k]$, we have $\alpha_{ij} \neq \alpha_{is}$, $\forall i \in [1, \dots, m]$ and $\forall j, s \in [1, \dots, k]$. Therefore, $\boldsymbol{t}_2 = \lambda \boldsymbol{t}_1$, $\lambda \in \mathbb{R}$. Since $\boldsymbol{t}_1, \boldsymbol{t}_2 \in [0, 1]^k$, $\sum_{i=1}^{k} t_{1i} = 1$ and $\sum_{i=1}^{k} t_{2i} = 1$, we have $\lambda = 1$, which indicates that $T_i(\boldsymbol{t}_1) = T_i(\boldsymbol{t}_2)$ if and only if $\boldsymbol{t}_1 = \boldsymbol{t}_2$. $\qquad\square$

## B.2  PROOF OF LEMMA 3.3

If the cross entropy loss is used, we have the following optimization problem:

$$\phi(g) = -\sum_{i=1}^{k} p(y = i|\boldsymbol{x}) \log(g_i(\boldsymbol{x}))$$

$$\text{s.t.} \quad \sum_{i=1}^{k} g_i(\boldsymbol{x}) = 1.$$

By using the Lagrange multiplier method, we can obtain the following non-constrained optimization problem:

$$\Phi(g) = -\sum_{i=1}^{k} p(y = i|\boldsymbol{x}) \log(g_i(\boldsymbol{x})) + \lambda \left( \sum_{i=1}^{k} g_i(\boldsymbol{x}) - 1 \right).$$

The derivative of $\Phi(g)$ with respect to $g$ is

$$\frac{\partial \Phi(g)}{\partial g} = \left[ -\frac{p(y = 1|\boldsymbol{x})}{g_1(\boldsymbol{x})} + \lambda, \dots, -\frac{p(y = k|\boldsymbol{x})}{g_k(\boldsymbol{x})} + \lambda \right]^\top.$$

By setting this derivative to 0 we obtain

$$g_i^\star(\boldsymbol{x}) = \frac{1}{\lambda} p(y = i | \boldsymbol{x}), \ \forall i \in [1, \ldots, k] \text{ and } \forall \boldsymbol{x} \in \mathcal{X}.$$

Since $\sum_{i=1}^k g_i^\star(\boldsymbol{x}) = 1$ and $\sum_{i=1}^k p(y = i | \boldsymbol{x}) = 1$, we have

$$\sum_{i=1}^k g_i^\star(\boldsymbol{x}) = \sum_{i=1}^k \frac{1}{\lambda} p(y = i | \boldsymbol{x}) = 1.$$

Therefore, we can easily obtain $\lambda = 1$. In this way, $g_i^\star(\boldsymbol{x}) = p(y = i | \boldsymbol{x})$, which concludes the proof. $\qquad\square$

## B.3 Proof of Theorem 3.4

According to Lemma 3.3, by minimizing $R_{\mathrm{ccm}}(f)$ with the cross entropy loss, we can obtain

$$q_i^\star(\boldsymbol{x}) = p(\bar{y} = i | \boldsymbol{x}), \ \forall i \in [1, \ldots, m].$$

Let us introduce $\bar{\boldsymbol{p}} = [p(\bar{y} = 1 | \boldsymbol{x}), \ldots, p(\bar{y} = m | \boldsymbol{x})]$ and $\boldsymbol{p} = [p(y = 1 | \boldsymbol{x}), \ldots, p(y) = k | \boldsymbol{x})]$. We have $\bar{\boldsymbol{p}} = \boldsymbol{T}(\boldsymbol{p})$. Since $q^\star(\boldsymbol{x}) = \bar{\boldsymbol{p}}$ and $g^\star(\boldsymbol{x}) = \boldsymbol{p}$, we have $q^\star(\boldsymbol{x}) = \boldsymbol{T}(g^\star(\boldsymbol{x}))$ where $g^\star(\boldsymbol{x}) = \mathrm{softmax}(f^\star(\boldsymbol{x}))$.

On the other hand, we can obtain $g_{\mathrm{ccm}}(\boldsymbol{x})$ by minimizing $R_{\mathrm{ccm}}(f)$ (i.e., $g_{\mathrm{ccm}}(\boldsymbol{x}) = \mathrm{softmax}(f_{\mathrm{ccm}}(\boldsymbol{x}))$). Then, we can obtain $q^\star(\boldsymbol{x}) = \boldsymbol{T}(g_{\mathrm{ccm}}(\boldsymbol{x}))$, which further ensures $\boldsymbol{T}(g^\star(\boldsymbol{x})) = \boldsymbol{T}(g_{\mathrm{ccm}}(\boldsymbol{x}))$. Therefore, according to Lemma 3.2, we obtain $g^\star(\boldsymbol{x}) = g_{\mathrm{ccm}}(\boldsymbol{x})$, i.e., $f^\star(\boldsymbol{x}) = f_{\mathrm{ccm}}(\boldsymbol{x})$. $\qquad\square$

## B.4 Proof of Theorem 3.5

We define a function space for our CC method as

$$\mathcal{G}_{\mathrm{ccm}} = \{g : (\boldsymbol{x}, \bar{y}) \to \mathcal{L}(\boldsymbol{T}(\boldsymbol{\eta}(\boldsymbol{x})), \bar{y}) \mid f \in \mathcal{F}\},$$

where

$$\mathcal{L}(\boldsymbol{T}(\boldsymbol{\eta}(\boldsymbol{x})), \bar{y}) = -\log\Big(\frac{\sum_{j=1}^k \rho_{\bar{y}} \alpha_{\bar{y}j} \exp(f_j(\boldsymbol{x}))}{\sum_{j=1}^k \beta_j \exp(f_j(\boldsymbol{x}))}\Big),$$

$f_j(\boldsymbol{x})$ denotes the $j$-th element of the output. Then we have the following lemmas.

**Lemma B.1.** *Let the loss function $\mathcal{L}(\boldsymbol{T}(\boldsymbol{\eta}(\boldsymbol{x})), \bar{y})$ be $L'$-Lipschitz with respect to $f(\boldsymbol{x})$ ($0 < L' < \infty$) and bounded by $C_\ell'$, i.e., $\sup_{\boldsymbol{x} \in \mathcal{X}, f \in \mathcal{F}, \bar{y} \in [m]} \mathcal{L}(\boldsymbol{T}(\boldsymbol{\eta}(\boldsymbol{x})), \bar{y}) \leqslant C_\ell'$. Then for any $\delta > 0$, with probability at least $1 - \delta$,*

$$\sup_{f \in \mathcal{F}} |\widehat{R}_{\mathrm{ccm}}(f) - R_{\mathrm{ccm}}(f)| \leqslant 2\sqrt{2} L' \sum_{y=1}^k \mathfrak{R}_n(\mathcal{H}_y) + C_\ell' \sqrt{\frac{\log \frac{2}{\delta}}{2n}}.$$

*Proof.* We first show that the one direction $\sup_{f \in \mathcal{F}} \widehat{R}_{\mathrm{ccm}}(f) - R_{\mathrm{ccm}}(f)$. The change of $\sup_{f \in \mathcal{F}} \widehat{R}_{\mathrm{ccm}}(f) - R_{\mathrm{ccm}}(f)$ is no greater than $\frac{C_\ell'}{n}$ if an example $\boldsymbol{x}_i$ is replaced with an arbitrary example $\boldsymbol{x}_i'$. By applying McDiarmid's inequality (McDiarmid, 1989), for any $\delta > 0$, with probability at least $1 - \frac{\delta}{2}$,

$$\sup_{f \in \mathcal{F}} \widehat{R}_{\mathrm{ccm}}(f) - R_{\mathrm{ccm}}(f) \leqslant \mathbb{E}[\sup_{f \in \mathcal{F}} \widehat{R}_{\mathrm{ccm}}(f) - R_{\mathrm{ccm}}(f)] + C_\ell' \sqrt{\frac{\log \frac{2}{\delta}}{2n}}.$$

Then it is routine (Mohri et al., 2012) to show

$$\mathbb{E}[\sup_{f \in \mathcal{F}} \widehat{R}_{\mathrm{ccm}}(f) - R_{\mathrm{ccm}}(f)] \leqslant 2\mathfrak{R}_n(\mathcal{G}_{\mathrm{ccm}})$$
$$\leqslant 2\sqrt{2} L' \sum_{y=1}^k \mathfrak{R}_n(\mathcal{H}_y),$$

where the last inequality is due to the vector-contraction inequality for Rademacher complexities (Maurer, 2016).

By further taking into account the other side $\sup_{f \in \mathcal{F}} R_{\mathrm{ccm}}(f) - \widehat{R}_{\mathrm{ccm}}(f)$, we have for any $\delta > 0$, with probability at least $1 - \delta$,

$$\sup_{f \in \mathcal{F}} |\widehat{R}_{\mathrm{ccm}}(f) - R_{\mathrm{ccm}}(f)| \leqslant 2\sqrt{2}L' \sum_{y=1}^{k} \mathfrak{R}_n(\mathcal{H}_y) + C'_\ell \sqrt{\frac{\log \frac{2}{\delta}}{2n}},$$

which concludes the proof. □

**Lemma B.2.** *Let $\hat{f}$ be the empirical minimizer (i.e., $\hat{f} = \arg\min_{f \in \mathcal{F}} \widehat{R}(f)$) and $f^\star$ be the true risk minimizer (i.e., $f^\star = \arg\min_{f \in \mathcal{F}} R(f)$), then the following inequality holds:*

$$R(\hat{f}) - R(f^\star) \leqslant 2 \sup_{f \in \mathcal{F}} |\widehat{R}(f) - R(f)|.$$

*Proof.* We have

$$R(\hat{f}) - R(f^\star) = \widehat{R}(\hat{f}) - \widehat{R}(f^\star) + R(\hat{f}) - \widehat{R}(\hat{f}) + \widehat{R}(f^\star) - R(f^\star)$$
$$\leqslant 0 + 2 \sup_{f \in \mathcal{F}} |\widehat{R}(f) - R(f)|,$$

which concludes the proof. □

By combining Lemma B.1 and Lemma B.2, Theorem 3.5 can be proved. □

## C  PROOF OF RISK-CONSISTENT METHOD

### C.1  PROOF OF THEOREM 3.6

The classification risk could be rewritten as:

$$\begin{aligned}
R(f) &= \mathbb{E}_{p(\boldsymbol{x}, y)}[\mathcal{L}(f(\boldsymbol{x}), y)] \\
&= \int_{\mathcal{X}} \sum_{j=1}^{k} p(y = j, \boldsymbol{x}) \mathcal{L}(f(\boldsymbol{x}), j) \, \mathrm{d}\boldsymbol{x} \\
&= \int_{\mathcal{X}} \sum_{j=1}^{k} \sum_{i=1}^{m} p(y = j, \bar{y} = i, \boldsymbol{x}) \mathcal{L}(f(\boldsymbol{x}), j) \mathrm{d}\boldsymbol{x} \\
&= \int_{\mathcal{X}} \sum_{i=1}^{m} \sum_{j=1}^{k} p(y = j, \bar{y} = i, \boldsymbol{x}) \mathcal{L}(f(\boldsymbol{x}), j) \mathrm{d}\boldsymbol{x} \\
&= \int_{\mathcal{X}} \sum_{i=1}^{m} p(\bar{y} = i, \boldsymbol{x}) \sum_{j=1}^{k} \frac{p(y = j, \bar{y} = i, \boldsymbol{x})}{p(\bar{y} = i, \boldsymbol{x})} \mathcal{L}(f(\boldsymbol{x}), j) \mathrm{d}\boldsymbol{x} \\
&= \int_{\mathcal{X}} \sum_{i=1}^{m} p(\bar{y} = i, \boldsymbol{x}) \sum_{j=1}^{k} p(y = j \mid \bar{y} = i, \boldsymbol{x}) \mathcal{L}(f(\boldsymbol{x}), j) \mathrm{d}\boldsymbol{x} \\
&= \mathbb{E}_{p(\boldsymbol{x}, \bar{y})} \Big[ \sum_{j=1}^{k} p(y = j \mid \bar{y}, \boldsymbol{x}) \mathcal{L}(f(\boldsymbol{x}), j) \Big] = R_{\mathrm{rcm}}(f).
\end{aligned} \tag{9}$$

### C.2  EM ALGORITHM

In this section, we show that the training process of RCM works in an expectation-maximization manner and aims to maximize the likelihood $p(\bar{y}, \boldsymbol{x}; \boldsymbol{\gamma})$ with respect to $\boldsymbol{\gamma}$ when the widely used cross entropy loss is employed. Here, $\boldsymbol{\gamma}$ denotes the parameters of the classification model and $p(\cdot; \boldsymbol{\gamma})$ denotes the probability function approximated by the classification model. $\omega_j^{(v)}$ is used to represent the weight corresponding to the $j$-th class for the $v$-th example, and $0 \leqslant \omega_j^{(i)} \leqslant 1$, $\sum_{j=1}^{k} \omega_j^{(v)} = 1$. Then, the following equations hold:

$$\sum_{v=1}^{n} \log(p(\bar{y}^{(v)}, \boldsymbol{x}^{(v)}; \boldsymbol{\gamma})) = \sum_{v=1}^{n} \log(\sum_{j=1}^{k} p(\bar{y}^{(v)}, y^{(v)} = j, \boldsymbol{x}^{(v)}; \boldsymbol{\gamma}))$$

$$= \sum_{v=1}^{n} \log(\sum_{j=1}^{k} \omega_j^{(v)} \frac{p(\bar{y}^{(v)}, y^{(v)} = j, \boldsymbol{x}^{(v)}; \boldsymbol{\gamma})}{\omega_j^{(v)}})$$

$$\geqslant \sum_{v=1}^{n} \sum_{j=1}^{k} \omega_j^{(v)} \log(\frac{p(\bar{y}^{(v)}, y^{(v)} = j, \boldsymbol{x}^{(v)}; \boldsymbol{\gamma})}{\omega_j^{(v)}}).$$

The last inequality is based on Jensen's inequality and the properties of the weights: $0 \leqslant \omega_j^{(i)} \leqslant 1$, $\sum_{j=1}^{k} \omega_j^{(v)} = 1$. The inequality holds equality when $\frac{p(\bar{y}^{(v)}, y^{(v)}=j, \boldsymbol{x}^{(v)}; \boldsymbol{\gamma})}{\omega_j^{(v)}}$ is a constant, which means $\frac{p(\bar{y}^{(v)}, y^{(v)}=j, \boldsymbol{x}^{(v)}; \boldsymbol{\gamma})}{\omega_j^{(v)}} = C$ and $C$ is a constant when $v$ is fixed. Then, we have

$$\frac{p(\bar{y}^{(v)}, y^{(v)=j}, \boldsymbol{x}^{(v)}; \boldsymbol{\gamma})}{C} = \omega_j^{(v)},$$

$$\sum_{j=1}^{k} \frac{p(\bar{y}^{(v)}, y^{(v)=j}, \boldsymbol{x}^{(v)}; \boldsymbol{\gamma})}{C} = \sum_{j=1}^{k} \omega_j^{(v)},$$

$$\frac{p(\bar{y}^{(v)}, \boldsymbol{x}^{(v)}; \boldsymbol{\gamma})}{C} = 1,$$

$$p(\bar{y}^{(v)}, \boldsymbol{x}^{(v)}; \boldsymbol{\gamma}) = C.$$

In this way, the value of $\omega_j^{(v)}$ could be calculated by

$$\omega_j^{(v)} = \frac{p(\bar{y}^{(v)}, y^{(v)} = j, \boldsymbol{x}^{(v)}; \boldsymbol{\gamma})}{C},$$

$$\omega_j^{(v)} = \frac{p(\bar{y}^{(v)}, y^{(v)} = j, \boldsymbol{x}^{(v)}; \boldsymbol{\gamma})}{p(\bar{y}^{(v)}, \boldsymbol{x}^{(v)}; \boldsymbol{\gamma})},$$

$$\omega_j^{(v)} = p(y^{(v)} = j \mid \bar{y}^{(v)}, \boldsymbol{x}^{(v)}; \boldsymbol{\gamma}).$$

Therefore, the E-step of RCM is to set $\omega_j^{(v)} = p(y^{(v)} = j \mid \bar{y}^{(v)}, \boldsymbol{x}^{(v)}; \boldsymbol{\gamma})$, to make the inequality holds with equality, which means to maximize $\sum_{v=1}^{n} \sum_{j=1}^{k} \omega_j^{(v)} \log(\frac{p(\bar{y}^{(v)}, y^{(v)}=j, \boldsymbol{x}^{(v)}; \boldsymbol{\gamma})}{\omega_j^{(v)}})$ with fixed $p(\bar{y}^{(v)}, y^{(v)} = j, \boldsymbol{x}^{(v)}; \boldsymbol{\gamma})$. For the M-step, RCM is to maximize $\sum_{v=1}^{n} \sum_{j=1}^{k} \omega_j^{(v)} \log(\frac{p(\bar{y}^{(v)}, y^{(v)}=j, \boldsymbol{x}^{(v)}; \boldsymbol{\gamma})}{\omega_j^{(v)}})$ with fixed $\omega_j^{(v)}$. We have the following equations with the fixed $\omega_j^{(v)}$:

$$\log(\frac{p(\bar{y}^{(v)}, y^{(v)} = j, \boldsymbol{x}^{(v)}; \boldsymbol{\gamma})}{\omega_j^{(v)}})$$

$$= \log(p(\bar{y}^{(v)}, y^{(v)} = j, \boldsymbol{x}^{(v)}; \boldsymbol{\gamma})) - \log(\omega_j^{(v)})$$

$$= \log(p(\boldsymbol{x} \mid \bar{y}^{(v)}, y^{(v)} = j; \boldsymbol{\gamma})) + \log(p(\bar{y}^{(v)}, y^{(v)} = j); \boldsymbol{\gamma}) - \log(\omega_j^{(v)})$$

$$= \log(p(\boldsymbol{x} \mid y^{(v)} = j; \boldsymbol{\gamma})) + \log(p(\bar{y}^{(v)}, y^{(v)} = j; \boldsymbol{\gamma})) - \log(\omega_j^{(v)})$$

$$= \log(\frac{p(y^{(v)} = j \mid \boldsymbol{x}; \boldsymbol{\gamma}) p(\boldsymbol{x}; \boldsymbol{\gamma})}{p(y^{(v)} = j; \boldsymbol{\gamma})}) + \log(p(\bar{y}^{(v)}, y^{(v)} = j); \boldsymbol{\gamma}) - \log(\omega_j^{(v)})$$

$$= \log(p(y^{(v)} = j \mid \boldsymbol{x}; \boldsymbol{\gamma})) + \log(\frac{p(\boldsymbol{x}; \boldsymbol{\gamma})}{p(y^{(v)} = j; \boldsymbol{\gamma})}) + \log(p(\bar{y}^{(v)}, y^{(v)} = j; \boldsymbol{\gamma})) - \log(\omega_j^{(v)})$$

It is noteworthy that $\log(\frac{p(\boldsymbol{x}; \boldsymbol{\gamma})}{p(y^{(v)}=j; \boldsymbol{\gamma})}) + \log(p(\bar{y}^{(v)}, y^{(v)} = j; \boldsymbol{\gamma})) - \log(\omega_j^{(v)})$ could be treated as a constant term with respect to $\boldsymbol{\gamma}$ (i.e., $\log(\frac{p(\boldsymbol{x}; \boldsymbol{\gamma})}{p(y^{(v)}=j; \boldsymbol{\gamma})}) + \log(p(\bar{y}^{(v)}, y^{(v)} = j; \boldsymbol{\gamma})) = \log(\frac{p(\boldsymbol{x})}{p(y^{(v)}=j)} +$

$\log(p(\bar{y}^{(v)}, y^{(v)} = j)))$ since the role of classification model is to approximate the probability function $p(y \mid \boldsymbol{x})$. When the cross entropy loss function is employed, $\mathcal{L}(f(\boldsymbol{x}), j) = -\log(p(y = j \mid \boldsymbol{x}); \boldsymbol{\gamma})$, which means maximizing $\log(p(y = j \mid \boldsymbol{x}); \boldsymbol{\gamma})$ is equivalent to minimizing $\mathcal{L}(f(\boldsymbol{x}), j)$.

In summary, the E-step of the RCM is to maximize the lower bound of $p(\bar{y}, \boldsymbol{x}; \boldsymbol{\gamma})$ and the M-step of RCM is to maximize the improved lower-bound by updating the parameters $\boldsymbol{\gamma}$ of the classification model.

Based on the above analysis, the M-step of RCM is to minimize $\sum_{v=1}^{n} \sum_{j=1}^{k} \omega_j^{(v)} \mathcal{L}(f(\boldsymbol{x}^{(v)}), j)$ with fixed $\omega_j^{(v)}$.

### C.3 Proof of Theorem 3.7

We define a function space for our RCM method as

$$\mathcal{G}_{\mathrm{rcm}} = \{g : (\boldsymbol{x}, \bar{y}) \to \sum_{j=1}^{k} p(y = j \mid \bar{y}, \boldsymbol{x}) \mathcal{L}(f(\boldsymbol{x}), j) \mid f \in \mathcal{F}\},$$

where $(\boldsymbol{x}, \bar{y})$ is randomly sampled from $\bar{p}(\boldsymbol{x}, \bar{y})$. Let $\mathfrak{R}_n(\mathcal{G}_{\mathrm{rcm}})$ be the expected Rademacher complexity of $\mathcal{G}_{\mathrm{rcm}}$. Then, to prove Theorem 3.7, we introduce the following lemmas.

**Lemma C.1.** *Let the used multi-class classification loss function $\mathcal{L}(f(\boldsymbol{x}), y)$ be bounded by $C_\ell$, i.e., $\sup_{\boldsymbol{x} \in \mathcal{X}, f \in \mathcal{F}, y \in \mathcal{Y}} \mathcal{L}(f(\boldsymbol{x}), y) \leqslant C_\ell$. Then, for any $\delta > 0$, with probability at least $1 - \delta$,*

$$\mathbb{E}[\hat{R}_{\mathrm{rcm}}(\hat{f}_{\mathrm{rcm}})] - \hat{R}_{\mathrm{rcm}}(\hat{f}_{\mathrm{rcm}}) \leqslant 2\mathfrak{R}_n(\mathcal{G}_{\mathrm{rcm}}) + C_\ell \sqrt{\frac{\log \frac{1}{\delta}}{2n}}.$$

*Proof.* Since $\sum_{j=1}^{k} p(y = j \mid \bar{y}, \boldsymbol{x}) = 1$ and $\mathcal{L}(f(\boldsymbol{x}), y)$ is bounded by $C_\ell$, the change of $\mathbb{E}[\hat{R}_{\mathrm{rcm}}(\hat{f}_{\mathrm{rcm}})] - \hat{R}_{\mathrm{rcm}}(\hat{f}_{\mathrm{rcm}})$ is no greater than $\frac{C_\ell}{n}$ if an example $\boldsymbol{x}_i$ is replaced with an arbitrary example $\boldsymbol{x}_i'$. By applying McDiarmid's inequality (McDiarmid, 1989), for any $\delta > 0$, with probability at least $1 - \delta$,

$$\mathbb{E}[\hat{R}_{\mathrm{rcm}}(\hat{f}_{\mathrm{rcm}})] - \hat{R}_{\mathrm{rcm}}(\hat{f}_{\mathrm{rcm}}) \leqslant \sup_{f \in \mathcal{F}} \mathbb{E}[\hat{R}_{\mathrm{rcm}}(f)] - \hat{R}_{\mathrm{rcm}}(f)$$

$$\leqslant \mathbb{E}[\sup_{f \in \mathcal{F}} \mathbb{E}[\hat{R}_{\mathrm{rcm}}(f)] - \hat{R}_{\mathrm{rcm}}(f)] + C_\ell \sqrt{\frac{\log \frac{1}{\delta}}{2n}}.$$

Then we can bound the expectation of the right-hand side as follows:

$$\mathbb{E}[\sup_{f \in \mathcal{F}} \mathbb{E}[\hat{R}_{\mathrm{rcm}}(f)] - \hat{R}_{\mathrm{rcm}}(f)] \leqslant 2\mathfrak{R}_n(\mathcal{G}_{\mathrm{rcm}}),$$

which concludes the proof. $\square$

**Lemma C.2.** *Let the loss function $\mathcal{L}(f(\boldsymbol{x}), y)$ be $L$-Lipschitz with respect to $f(\boldsymbol{x})$ $(0 < L < \infty)$. The following inequality holds:*

$$\mathfrak{R}_n(\mathcal{G}_{\mathrm{rcm}}) \leqslant \sqrt{2}L \sum_{y=1}^{k} \mathfrak{R}_n(\mathcal{H}_y),$$

*where*

$$\mathcal{H}_y = \{h : \boldsymbol{x} \to f_y(\boldsymbol{x}) | f \in \mathcal{F}\},$$

$$\mathfrak{R}_n(\mathcal{H}_y) = \mathbb{E}_{\boldsymbol{\sigma}, \mathcal{X}}[\sup_{h \in \mathcal{H}_y} \frac{1}{n} \sum_{i=1}^{n} \sigma_i h(\boldsymbol{x})].$$

*Proof.* Since $\sum_{j=1}^{k} p(y = j \mid \bar{y}, \boldsymbol{x}) = 1$ and $0 \leqslant p(y = j \mid \bar{y}, \boldsymbol{x}) \leqslant 1$ $\forall y \in \mathcal{Y}$, we can obtain $\mathfrak{R}_n(\mathcal{G}_{\mathrm{rcm}}) \leqslant \mathfrak{R}_n(\mathcal{L} \circ \mathcal{F})$ where $\mathcal{L} \circ \mathcal{F}$ denotes $\{\mathcal{L} \circ f \mid f \in \mathcal{F}\}$. Since $\mathcal{H}_y = \{h : \boldsymbol{x} \to f_y(\boldsymbol{x}) \mid f \in \mathcal{F}\}$ and the loss function is $L$-Lipschitz, by using the vector contraction inequality for Rademacher complexities (Maurer, 2016), we have $\mathfrak{R}_n(\mathcal{L} \circ \mathcal{F}) \leqslant \sqrt{2}L \sum_{y=1}^{k} \mathfrak{R}_n(\mathcal{H}_y)$, which concludes the proof of Lemma C.2. $\square$

By combining Lemma C.1 and Lemma C.2, Theorem 3.7 can be proved. $\square$

## D DETAILED DESCRIPTIONS/EXPLANATIONS OF RCM AND CCM

In our paper, we present two loss functions to solve the MCMU problem: RCM and CCM. These two methods stem from different conceptual frameworks.

The main idea of CCM is to design a loss function by converting the problem of classifying multiple classes into multiple unlabeled datasets. This conversion is based on our built connection relationship between $p(y|\boldsymbol{x})$ (the probability that an instance $\boldsymbol{x}$ belongs to a label) and $p(\bar{y}|x)$ (the probability that an instance $\boldsymbol{x}$ belongs to an unlabeled set) in Eq. (2), which is further represented as a transition function $\boldsymbol{T}$ (i.e., $\bar{\boldsymbol{\eta}}(\boldsymbol{x}) = \boldsymbol{T}(\boldsymbol{\eta}(\boldsymbol{x}))$ where $\bar{\boldsymbol{\eta}}(\boldsymbol{x}) = p(\bar{y}|\boldsymbol{x})$ and $\boldsymbol{\eta}(\boldsymbol{x}) = p(y|\boldsymbol{x})$). As we aim to approximate $p(\bar{y}|\boldsymbol{x})$ by $T(g(\boldsymbol{x}))$, we can infer that $p(y|\boldsymbol{x})$ can be approximated by $g(\boldsymbol{x})$ (where we use $g(\boldsymbol{x})$ to denote the Softmax output of the model), because $\boldsymbol{T}$ is an injective function. The detailed proof can be found in Appendix B.3.

The main idea of RCM is risk rewriting. We rewrite the risk function on clean data (Eq. (1)) into an equivalent form for MCMU (Eq. (5)) that can be accessible from unlabeled sets, since the ordinary classification risk cannot be obtained from unlabeled sets. Concretely, we rewrite the risk function via the importance-weighting strategy, i.e. assign the weight $p(y = j|\boldsymbol{x})$ to loss $\mathcal{L}(f(\boldsymbol{x}), j)$. RCM could recover the distribution of clean data by combining the information of unlabeled sets and the learned classifier. Then we could achieve the optimization of Eq (1) by minimizing Eq. (5).

## E DETAILED INFORMATION OF EXPERIMENTS

### E.1 TRAINING DETAILS

In the experiments, Adam (Kingma & Ba, 2015) was used for optimization and we adopt the cross entropy function as the loss function for CCM and adopt softmax cross entropy for RCM. We ran 5 trials on each dataset for each method and recorded mean accuracy and standard deviation (mean±std). We trained the classification model for 100 epochs on all datasets. We recorded the average test accuracy of the last ten epochs as the accuracy for each trial except for U-Stop, in which we recorded the test accuracy of the epoch where stop training as the accuracy for each trial. The learning rate was chosen from $\{10^{-5}, 10^{-4}, 10^{-3}, 10^{-2}, 10^{-1}\}$ and the batch size was chosen from 128 and 256. The weight decay was set as $10^{-5}$. The hyper-parameters for compared methods were searched according to the suggested parameters by respective papers. We used PyTorch (Paszke et al., 2019) to implement our experiments and conducted the experiments on NVIDIA 3090 GPUs.

### E.2 COMPARED METHODS

In this section, we provide detailed descriptions of the compared methods.

- **Unbiased (Tang et al., 2022):** This method aims to minimize an unbiased risk estimator derived from by using the backward correction technique (Patrini et al., 2017). However, this method suffers from the unreasonable training objective caused by the negative empirical risk.

- **U-Stop (Tang et al., 2022):** It aims to minimize an identical unbiased risk estimator as the Unbiased method but stops training when the empirical risk goes negative to prevent suffering from the optimization obstacle caused by unbounded losses.

- **U-Correct (Tang et al., 2022):** It takes the absolute value of partial risks corresponding to each class in the Unbiased method to correct the partial risks to be non-negative.

- **U-Flood (Tang et al., 2022):** It aims to apply the flooding method (Ishida et al., 2020) to the Unbiased method to maintain the empirical risks at certain levels.

- **Prop (Yu et al., 2014):** It treats class priors as weak supervision to the set and minimizes the difference between the true class priors and the class priors predicted by the classification model. KL divergence is applied to measure the difference.

- **U-PRR (Tang et al., 2022):** It aims to utilize a partial risk regularization that maintains the partial risks corresponding to each class in the Unbiased method to certain levels.

# F  ADDITIONAL EXPERIMENTS

## F.1  ROBUSTNESS AGAINST NOISY PRIOR MATRIX

We conducted experiments that add noise to the true class prior matrix $\Theta$ on $m = 10$ ($m$ denotes the number of sets). Specifically, we train the classifier on the MNIST dataset and the class prior matrix we used is the random matrix as we described in Section 4.1. We design two procedures to perturb the class prior matrix with the noise rate $\epsilon$ and the class number $k$.

- Random noise: we obtain the noisy class prior matrix by right-multiplying the truth class prior matrix with a noise matrix $\boldsymbol{N}$, i.e., $\Theta_{\text{noise}} = \Theta \cdot \boldsymbol{N}$ where $N_{ii} = 1 - \frac{k-1}{k}\epsilon$ and $N_{ij} = \frac{\epsilon}{k}$ where $i \neq j$. This noise procedure typically only changes the numerical values of the elements of the prior matrix, without significantly altering the relative order of magnitude between elements in the same row. The experimental results are reported in Table 4.

- Order disturbance: we aim to disrupt the relative order of class priors in the same row. For example, given the $i$-th row of a prior matrix be $[0.1, 0.2, 0.7]$ and the relative order is $\theta_{i3} > \theta_{i2} > \theta_{i1}$. The noisy version $[0.12, 0.18, 0.7]$ only changes the numerical values while the noisy version $[0.18, 0.12, 0.7]$ changes both numerical values and the relative order (i.e., changes the relative order to $\theta_{i3} > \theta_{i1} > \theta_{i2}$). Concretely, we perturb to the $i$-th row of $\Theta$ by the following step. Firstly, we generate an arithmetic sequence $\boldsymbol{l}$ according to $\epsilon$, i.e., $\boldsymbol{l} = \{1, 1 + \epsilon, \ldots, 1 + (k-1)\epsilon\}$. Secondly, we randomly shuffle the elements in $\boldsymbol{l}$. Thirdly, we divide the elements in the ith row by the corresponding element in $\boldsymbol{l}$, i.e. $\theta_{ij} = \theta_{ij}/l_j$. Finally, we normalize the $i$-th row, i.e., $\theta_{ij} = \theta_{ij}/\sum_{j=1}^{k} \theta_{ij}$. The experimental results of this case are reported in Table 5.

To our surprise, in Table 4, all methods other than CCM works well and were not significantly affected by the random noise. Additionally, our proposed RCM model achieves the best performance on the inaccurate prior matrix. We observe that CCM tends to overfit in cases of high noise rates, which leads to poorer performance.

From Table 5, we can observe in order disturbance, as the noise rate increases, the performance of all models decreases. However, our proposed RCM method still achieves the best performance in almost all cases.

By comparing the experimental results on random noise and order disturbance, it appears that the inaccurate elements in the prior matrix may not have a substantial impact on the performance of methods. What appears to be a more influential factor is the disturbance of the relative order of the prior matrix. In addition, RCM performs well even in the presence of noise in the prior matrix. This can be attributed to the label correction effect of the importance weighting scheme in RCM, which could leverage the model's capability to refine the labeling information in surrogate labels.

Table 4: Classification performance of each method on the MNIST dataset with $m = 10$ trained on random prior matrix with random noise

| Noise Rate | 0 | 0.1 | 2 | 0.3 | 0.5 | 0.7 |
|---|---|---|---|---|---|---|
| Unbiased | $14.10 \pm 2.94\%$ | $12.89 \pm 1.89\%$ | $14.58 \pm 0.91\%$ | $15.52 \pm 1.28\%$ | $18.68 \pm 1.90\%$ | $19.16 \pm 0.98\%$ |
| U-Stop | $25.39 \pm 5.52\%$ | $20.49 \pm 6.58\%$ | $18.99 \pm 3.80\%$ | $26.73 \pm 5.84\%$ | $21.27 \pm 4.10\%$ | $24.72 \pm 6.98\%$ |
| U-Corret | $27.32 \pm 7.07\%$ | $27.29 \pm 5.95\%$ | $27.15 \pm 6.37\%$ | $26.70 \pm 6.20\%$ | $26.78 \pm 6.97\%$ | $25.48 \pm 6.32\%$ |
| U-Flood | $86.83 \pm 1.31\%$ | $88.46 \pm 1.21\%$ | $86.84 \pm 2.43\%$ | $85.87 \pm 1.51\%$ | $75.08 \pm 3.05\%$ | $62.39 \pm 6.99\%$ |
| Prop | $88.64 \pm 1.93\%$ | $88.85 \pm 1.50\%$ | $89.78 \pm 1.48\%$ | $89.66 \pm 0.84\%$ | $88.99 \pm 0.92\%$ | $87.88 \pm 1.49\%$ |
| U-PRR | $23.17 \pm 1.59\%$ | $23.86 \pm 1.75\%$ | $25.57 \pm 2.75\%$ | $26.10 \pm 2.26\%$ | $24.61 \pm 2.06\%$ | $22.19 \pm 1.79\%$ |
| CCM | $91.46 \pm 1.22\%$ | $91.11 \pm 1.56\%$ | $37.49 \pm 37.88\%$ | $10.31 \pm 1.45\%$ | $43.15 \pm 23.79\%$ | $11.68 \pm 1.82\%$ |
| RCM | $94.80 \pm 0.64\%$ | $95.25 \pm 0.58\%$ | $95.41 \pm 0.53\%$ | $95.56 \pm 0.48\%$ | $95.45 \pm 0.53\%$ | $91.92 \pm 4.69\%$ |

## F.2  ROBUSTNESS AGAINST VARIANT SET NUMBERS

We conducted additional experiments with different numbers of sets with fixed data points (i.e. with fixed set number times the set size) on the MNIST dataset. Random matrix and asymmetric matrix are adapted as the class prior matrix. The experimental results are shown in Table 6 and Table 7. Our proposed RCM and CCM demonstrate stable performance with the increase in the number

Table 5: Classification performance of each method on the MNIST dataset with $m = 10$ trained on random prior matrix with order disturbance

| Noise Rate | 0 | 0.1 | 0.2 | 0.3 | 0.5 | 0.7 |
|---|---|---|---|---|---|---|
| Unbiased | $14.10 \pm 2.94\%$ | $17.22 \pm 4.02\%$ | $15.84 \pm 3.00\%$ | $9.87 \pm 7.11\%$ | $9.87 \pm 7.11\%$ | $8.34 \pm 1.86\%$ |
| U-Stop | $25.39 \pm 5.52\%$ | $19.97 \pm 1.96\%$ | $20.61 \pm 3.12\%$ | $14.02 \pm 3.36\%$ | $5.08 \pm 1.30\%$ | $8.52 \pm 4.24\%$ |
| U-Correct | $27.32 \pm 7.07\%$ | $30.18 \pm 6.56\%$ | $18.63 \pm 5.92\%$ | $14.01 \pm 2.87\%$ | $15.47 \pm 4.76\%$ | $15.23 \pm 3.78\%$ |
| U-Flood | $86.83 \pm 1.31\%$ | $87.70 \pm 0.79\%$ | $85.06 \pm 0.34\%$ | $58.28 \pm 28.31\%$ | $54.10 \pm 31.16\%$ | $55.00 \pm 19.91\%$ |
| Prop | $88.64 \pm 1.93\%$ | $86.98 \pm 2.06\%$ | $80.55 \pm 3.90\%$ | $73.98 \pm 1.39\%$ | $61.53 \pm 3.71\%$ | $50.93 \pm 5.51\%$ |
| U-PRR | $23.17 \pm 1.59\%$ | $21.92 \pm 2.60\%$ | $20.55 \pm 2.13\%$ | $14.12 \pm 3.43\%$ | $9.96 \pm 4.16\%$ | $11.32 \pm 4.26\%$ |
| CCM | $91.46 \pm 1.22\%$ | $89.27 \pm 2.12\%$ | $86.99 \pm 2.62\%$ | $82.03 \pm 0.88\%$ | $74.84 \pm 4.57\%$ | $68.91 \pm 5.94\%$ |
| RCM | $94.80 \pm 0.64\%$ | $93.09 \pm 1.98\%$ | $87.50 \pm 1.51\%$ | $83.78 \pm 1.28\%$ | $75.83 \pm 5.25\%$ | $67.69 \pm 7.02\%$ |

of sets and still outperform the baseline methods. When the number of sets becomes excessively large, some baseline methods may achieve poorer performance. This is because in such a case, the distribution $p(x)$ on training data might shift from testing data. Some papers Zhang et al. (2020a); Shimodaira (2000) refer to this case as a covariate shift, which would lead to degraded performance.

Table 6: Classification performance of each method on the MNIST dataset using an asymmetric prior matrix with variant set numbers.

| Methods | m=30 | m=50 | m=100 | m=200 | m=300 | m=500 | m=1000 |
|---|---|---|---|---|---|---|---|
| Unbiased | 84.30±2.55% | 86.85±1.06% | 86.74±0.88% | 87.75±0.47% | 88.17±0.68% | 87.46±0.41% | 62.78±5.07% |
| U-Stop | 89.69±1.21% | 89.79±1.59% | 91.05±0.42% | 89.47±0.20% | 89.71±0.58% | 88.37±0.21% | 85.23±0.65% |
| U-Correct | 94.07±0.18% | 94.16±0.23% | 87.00±0.30% | 78.94±1.86% | 63.33±16.18% | 27.04±8.39% | 26.80±7.59% |
| U-Flood | 93.42±0.56% | 93.29±0.34% | 90.55±0.53% | 85.29±0.86% | 79.46±0.83% | 68.43±10.75% | 56.70±3.33% |
| Prop | 82.98±1.55% | 79.85±1.01% | 75.46±1.23% | 72.77±1.60% | 72.62±0.67% | 71.61±0.51% | 71.90±0.32% |
| U-PRR | 91.11±0.29% | 89.32±0.89% | 79.28±1.34% | 70.88±2.36% | 67.80±0.69% | 62.76±2.53% | 67.28±2.11% |
| CCM | 95.77±0.12% | 95.66±0.17% | 95.66±0.08% | 95.40±0.14% | 95.69±0.14% | 95.57±0.20% | 95.91±0.17% |
| RCM | 95.94±0.23% | 96.02±0.08% | 95.95±0.15% | 95.70±0.23% | 95.88±0.08% | 95.82±0.07% | 95.77±0.22% |

Table 7: Classification performance of each method on the MNIST dataset using a random prior matrix with variant set numbers.

| Methods | m=30 | m=50 | m=100 | m=200 | m=300 | m=500 | m=1000 |
|---|---|---|---|---|---|---|---|
| Unbiased | 14.14±1.15% | 15.64±1.48% | 13.90±0.70% | 11.18±2.30% | 15.12±2.64% | 16.36±1.53% | 13.31±2.95% |
| U-Stop | 53.12±3.76% | 71.61±0.52% | 38.41±18.44% | 40.62±24.60% | 18.33±6.65% | 18.27±2.32% | 25.89±3.77% |
| U-Correct | 63.84±4.61% | 66.64±1.21% | 75.00±0.50% | 64.24±0.34% | 64.85±1.51% | 71.59±0.20% | 77.01±0.83% |
| U-Flood | 89.14±0.96% | 88.91±0.92% | 85.87±0.42% | 75.59±0.72% | 69.92±1.47% | 59.82±2.38% | 50.54±5.46% |
| Prop | 81.84±1.36% | 77.29±1.51% | 66.61±2.35% | 57.96±1.47% | 54.38±1.12% | 53.38±1.26% | 51.44±1.16% |
| U-PRR | 23.92±0.56% | 19.62±1.15% | 19.30±1.30% | 18.15±1.22% | 14.97±0.47% | 17.98±1.16% | 13.69±0.53% |
| CCM | 92.90±0.29% | 92.49±0.14% | 92.86±0.39% | 92.37±0.20% | 93.23±0.37% | 93.00±0.08% | 93.06±0.20% |
| RCM | 95.69±0.25% | 95.63±0.12% | 95.49±0.17% | 95.38±0.18% | 95.49±0.07% | 95.46±0.15% | 95.78±0.14% |

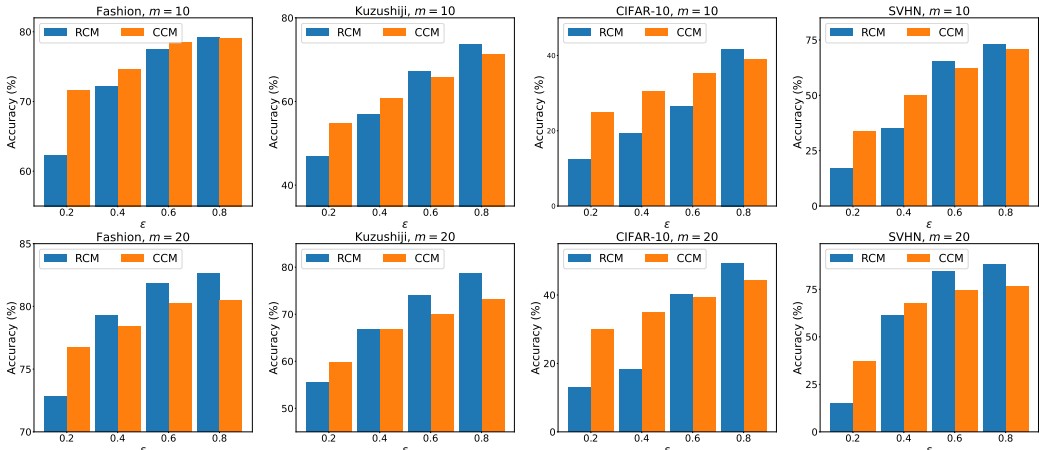

Figure 1: Mean accuracy over 3 trials in percentage for our proposed RCM and CCM tested on different set sizes on the random class prior matrix.

