# OpenReview forum: "Consistent Multi-Class Classification from Multiple Unlabeled Datasets"
_ICLR.cc/2024/Conference — ICLR 2024 spotlight_

### Official Review · Reviewer_XSqD · 2023-10-23

**Soundness:** 3 good
**Presentation:** 3 good
**Contribution:** 3 good
**Rating:** 6
**Confidence:** 4

**Summary:**

The paper introduces a novel approach to weakly supervised learning, specifically targeting multi-class classification from multiple unlabeled datasets with class priors. It addresses limitations identified in Tang et al., 2022, and offers two distinct approaches: a classifier-consistent method using a probability transition matrix and a risk-consistent approach employing importance weighting. However, certain areas require further clarification and empirical investigation.

**Strengths:**

- The paper successfully addresses limitations observed in the MCMU method and presents an alternative solution, expanding the landscape of weakly supervised learning.
- The introduction of a novel approach focusing on empirical risk minimization (ERM) at the instance level, treating individual data points, stands out as a significant strength.

**Weaknesses:**

- The experimental comparison is somewhat limited, primarily featuring methods from Tang et al., 2022. While justified due to MCMU's novelty, including additional weakly supervised methods could provide valuable reference points. Moreover, presenting accuracies for fully supervised learning on the same train/test splits would enhance the evaluation.
- The paper lacks a comprehensive study of the method's limitations, particularly with regards to constraints on parameters such as m (the number of unlabeled sets) and n_i (the number of data points in each set), which are crucial for practical applicability.
- Theoretical results like Theorem 3.5, which establishes an upper bound for the difference between true and empirical risk, would benefit from discussion regarding their practical implications and their relationship to the actual classification task involving class labels.

**Questions:**

1. The paper could further investigate the impact of small m (few unlabeled sets) and significant variations in label proportions across sets on the generalizability of its results with respect to m and n_i.

2. A clarification regarding the difference between Lemma 3.1 in the paper and Theorem 1 in Lu et al., ICML 2021, would be valuable for readers.

3. Given that label proportions in real-world scenarios may only slightly differ, the paper should outline constraints on parameters m and n for practical significance, especially considering that a large m may be necessary for the method's effectiveness.

4. A discussion on the practical relevance of the derived upper bound in Theorem 3.5 and its connection to the actual classification task involving class labels would provide valuable insights.

5. The observation that the proposed approach performs significantly worse under certain settings, such as a random class prior matrix and non-constrained m, should be supported by clearer details on how test accuracies were computed, whether test data were balanced, and if class size variations in datasets played a role.

Other suggestions:

"In MCMU, given the class priors, the data points in the same set are independent from each other, while in LLP, given the label proportions, the data points in
the same set are dependent from each other."

The second part of the sentence should clearly state whether data points are "independent from each other" or "dependent on each other." If it implies independence, it raises a significant ambiguity regarding how MCMU and LLP fundamentally differ, as class priors and label proportions are practically equivalent concepts. Moreover, there appears to be a conceptual misuse in the paragraph. Data points within each class are typically assumed to be independent, conditioned on their class labels. The statement in question uses "class priors" in place of "class labels", which could introduce confusion and should be clarified. Overall, given that label proportions and class priors are practically the same thing in the context studied by the papers it is hard to see any difference from a dependence perspective between the two approaches.

I think using the term "label proportions" when referring to class priors within each set would be more appropriate. In fact what the paper denotes as the j-th class prior of the i-th unlabeled set can be denoted as the label proportion of class j in dataset i. This would distinguish label proportions from class priors estimated using data from all available unlabeled sets.


In Section 4.1. "The ni data points contained in i-th unlabeled set were randomly generated according to ..." Isn't this supposed to be "randomly sampled" or "drawn".  Data already exists... nothing is supposed to be generated.

It would greatly enhance the paper if, within the related work section, an early explanation were provided regarding the reasons behind the possibility of negative empirical risk in these methods (Lu et al., 2019; Tsai & Lin, 2020; Tang et al., 2022). This proactive approach would help readers anticipate and better understand the issues related to negative empirical risk that the paper seeks to address.

In page 4 i is used for indexing both the sets and data points. It may cause confusions.

In Theorem 3.6 the paper defines this probability p(y=j|\bar{y},x)as the probability of x in \bar{y}-th unlabeled set whose ground-truth label is j. Isn't this supposed to be the probability of x in \bar{y}-th unlabeled set belonging to class j? x may or may not belong to class j. x's ground truth label is not necessarily j.

Theorem 3.6 also considers p(x|y=j)=p(x|y=j,\bar{y}) as a fact. However, this is not necessarily true. The first one is the conditional distribution of class j estimated using all data belonging to class j whereas the second one is the conditional distribution of class j estimated using only data from unlabeled set \bar{y}. So, from an empirical standpoint these distributions are not equal. Please clarify.


After Rebuttal:

Thanks for taking the time to do those extra comparisons I mentioned. It’s helpful to see how your method holds up with different 'm' values and varying levels of class imbalance. Based on these new results,  I'm happy to bump up my score by one.

---

> ### Author Response · Authors · 2023-11-19
> **Reply to Reviewer XSqD  (Part 1/4)**
>
> We sincerely thank you for spending so much time reviewing our paper! We are really touched by your strong sense of responsibility and meticulosity! We also appreciate the valuable suggestions you provided for our paper, which can definitely improve the quality of our paper! Our point-by-point responses are provided below.
>
> **Weakness 1: The experimental comparison is somewhat limited, primarily featuring methods from Tang et al., 2022. While justified due to MCMU's novelty, including additional weakly supervised methods could provide valuable reference points.**
>
> MCMU is a relatively new research area that was introduced very recently. To the best of our knowledge, there is only one work [1] that has studied this problem and this work proposes a number of effective methods to solve this MCMU problem. Therefore, we adopted all the methods in [1] as our baselines. We believe that the number of compared methods is sufficient, as we have already compared all the existing methods designed for MCMU. We would be very grateful if you could provide us with more weakly supervised learning methods that can be used for MCMU, and we will definitely include such methods in experimental comparisons.
>
> [1] Tang Y, Lu N, Zhang T, et al. Learning from Multiple Unlabeled Datasets with Partial Risk Regularization. ACML 2022.
>
> **Presenting accuracies for fully supervised learning on the same train/test splits would enhance the evaluation.**
>
> Thank you for your advice. We have conducted experiments with fully supervised data. The experimental results have been incorporated into Table 2 and Table 3 for comprehensive evaluations.
>
> **Weakness 2: The paper lacks a comprehensive study of the method's limitations, particularly with regards to constraints on parameters such as $m$ (the number of unlabeled sets) and $n_i$ (the number of data points in each set), which are crucial for practical applicability.**
>
> We conducted experiments on variants $m$ and $n_i$ and showed the experimental results in Appendix E. Here we provide a detailed description of the additional experimental results.
>
> We conducted additional experiments with different numbers of sets with fixed data points (i.e. with fixed set number times the set size) on the MNIST dataset. Random matrix and asymmetric matrix are adapted as the class prior matrix. The experimental results are shown in Table 1, Table 2, and Table 3. Our proposed RCM and CCM demonstrate stable performance with the increase in the number of sets and still outperform the baseline methods. When the number of sets becomes excessively large, some baseline methods may achieve poorer performance. This is because in such a case, the distribution $p(x)$ on training data might shift from testing data. Some papers [2,3] refer to this case as a covariate shift, which would lead to degraded performance.
>
> Table 1: Classification performance of each method on the MNIST dataset using an asymmetric prior matrix with variant set numbers.
>
> | Methods   | m=30         | m=50         | m=100        | m=200        | m=300         | m=500         | m=1000       |
> | --------- | ------------ | ------------ | ------------ | ------------ | ------------- | ------------- | ------------ |
> | Unbiased  | 84.30±2.55% | 86.85±1.06% | 86.74±0.88% | 87.75±0.47% | 88.17±0.68%  | 87.46±0.41%  | 62.78±5.07% |
> | U-Stop    | 89.69±1.21% | 89.79±1.59% | 91.05±0.42% | 89.47±0.20% | 89.71±0.58%  | 88.37±0.21%  | 85.23±0.65% |
> | U-Correct | 94.07±0.18% | 94.16±0.23% | 87.00±0.30% | 78.94±1.86% | 63.33±16.18% | 27.04±8.39%  | 26.80±7.59% |
> | U-Flood   | 93.42±0.56% | 93.29±0.34% | 90.55±0.53% | 85.29±0.86% | 79.46±0.83%  | 68.43±10.75% | 56.70±3.33% |
> | Prop      | 82.98±1.55% | 79.85±1.01% | 75.46±1.23% | 72.77±1.60% | 72.62±0.67%  | 71.61±0.51%  | 71.90±0.32% |
> | U-PRR     | 91.11±0.29% | 89.32±0.89% | 79.28±1.34% | 70.88±2.36% | 67.80±0.69%  | 62.76±2.53%  | 67.28±2.11% |
> | CCM       | 95.77±0.12% | 95.66±0.17% | 95.66±0.08% | 95.40±0.14% | 95.69±0.14%  | 95.57±0.20%  | **95.91±0.17%** |
> | RCM       | **95.94±0.23%** | **96.02±0.08%** | **95.95±0.15%** | **95.70±0.23%** | **95.88±0.08%**  | **95.82±0.07%**  | 95.77±0.22% |
>
> Continue with the next post

---

> ### Author Response · Authors · 2023-11-19
> **Reply to Reviewer XSqD  (Part 2/4)**
>
> Table 2: Classification performance of each method on the MNIST dataset using a random prior matrix with variant set numbers.
> | Methods   | m=30         | m=50         | m=100         | m=200         | m=300        | m=500        | m=1000       |
> | --------- | ------------ | ------------ | ------------- | ------------- | ------------ | ------------ | ------------ |
> | Unbiased  | 14.14±1.15% | 15.64±1.48% | 13.90±0.70%  | 11.18±2.30%  | 15.12±2.64% | 16.36±1.53% | 13.31±2.95% |
> | U-Stop    | 53.12±3.76% | 71.61±0.52% | 38.41±18.44% | 40.62±24.60% | 18.33±6.65% | 18.27±2.32% | 25.89±3.77% |
> | U-Correct | 63.84±4.61% | 66.64±1.21% | 75.00±0.50%  | 64.24±0.34%  | 64.85±1.51% | 71.59±0.20% | 77.01±0.83% |
> | U-Flood   | 89.14±0.96% | 88.91±0.92% | 85.87±0.42%  | 75.59±0.72%  | 69.92±1.47% | 59.82±2.38% | 50.54±5.46% |
> | Prop      | 81.84±1.36% | 77.29±1.51% | 66.61±2.35%  | 57.96±1.47%  | 54.38±1.12% | 53.38±1.26% | 51.44±1.16% |
> | U-PRR     | 23.92±0.56% | 19.62±1.15% | 19.30±1.30%  | 18.15±1.22%  | 14.97±0.47% | 17.98±1.16% | 13.69±0.53% |
> | CCM       | 92.90±0.29% | 92.49±0.14% | 92.86±0.39%  | 92.37±0.20%  | 93.23±0.37% | 93.00±0.08% | 93.06±0.20% |
> | RCM       | **95.69±0.25%** | **95.63±0.12%** | **95.49±0.17%**  | **95.38±0.18%**  | **95.49±0.07%** | **95.46±0.15%** | **95.78±0.14%** |
>
> Table 3: Classification performance of each method on the MNIST dataset using a random prior matrix with variant set numbers. ($m \geq k$)
>
> | Methods | Unbiased        | U-Stop           | U-Correct       | U-Flood         | Prop            | U-PRR           | CCM             | RCM             |
> | ------- | --------------- | ---------------- | --------------- | --------------- | --------------- | --------------- | --------------- | --------------- |
> | m=5     | 18.23$\pm$8.08% | 45.88$\pm$12.29% | 34.71$\pm$7.20% | 58.97$\pm$4.25% | 72.48$\pm$2.37% | 26.34$\pm$6.46% | 88.80$\pm$2.43% | **89.16$\pm$4.29%** |
> | m=6     | 14.61$\pm$6.72% | 42.02$\pm$22.68% | 40.06$\pm$8.00% | 68.00$\pm$3.39% | 82.06$\pm$4.49% | 28.45$\pm$5.02% | 89.80$\pm$3.59% | **92.71$\pm$0.73%** |
> | m=7     | 15.56$\pm$2.38% | 42.39$\pm$6.74%  | 49.72$\pm$8.40% | 77.53$\pm$7.95% | 89.32$\pm$3.43% | 32.78$\pm$5.33% | 90.66$\pm$2.19% | **94.53$\pm$1.06%** |
> | m=8     | 15.13$\pm$0.87% | 37.08$\pm$16.58% | 32.65$\pm$2.24% | 79.89$\pm$4.45% | 88.57$\pm$3.11% | 27.72$\pm$0.64% | 89.99$\pm$2.75% | **94.49$\pm$1.06%** |
> | m=9     | 14.06$\pm$1.79% | 27.77$\pm$8.25%  | 33.80$\pm$1.88% | 82.83$\pm$2.78% | 88.31$\pm$3.68% | 29.79$\pm$4.22% | 91.42$\pm$1.26% | **94.44$\pm$0.91%** |
>
> [2] Yiyang Zhang, Feng Liu, Zhen Fang, Bo Yuan, Guangquan Zhang, and Jie Lu. Clarinet: A one-step approach towards budget-friendly unsupervised domain adaptation. IJCAI 2020.
>
> [3] Hidetoshi Shimodaira. Improving predictive inference under covariate shift by weighting the log-likelihood function. Journal of Statistical Planning and Inference, 90(2):227–244, 2000.
>
> **Weakness 3: Theoretical results like Theorem 3.5, which establishes an upper bound for the difference between true and empirical risk, would benefit from discussion regarding their practical implications and their relationship to the actual classification task involving class labels.**
>
> The practical implication of Theorem 3.5 lies in the fact that the performance of our method can be improved, by increasing the training data of MCMU. We can also find that the optimal classifier obtained by our method will converge to the optimal classifier (learned by minimizing the actual classification risk involving class labels), as the number of training data of MCMU increases.
>
> **Q1: The paper could further investigate the impact of small m (few unlabeled sets) and significant variations in label proportions across sets on the generalizability of its results with respect to $m$ and $n_i$.**
>
> Thank you for your valuable suggestion. We have conducted experiments specifically designed for the case of small values of $m$. The experimental results are reported in Table 3. Please refer to our response to Weakness 2.
>
> **Q2: A clarification regarding the difference between Lemma 3.1 in the paper and Theorem 1 in Lu et al., ICML 2021, would be valuable for readers.**
>
> Lemma 3.1 can be considered as a multi-class extension of Theorem 1 (only for binary classification) in Lu et al., ICML 2021. As we indicated in our paper, CCM is a multi-class extension of Lu et al., ICML 2021. We would like to explain that the CCM is not the main contribution of this paper. The main contribution of our paper lies in the proposed RCM.
>
> **Q3: Given that label proportions in real-world scenarios may only slightly differ, the paper should outline constraints on parameters m and n for practical significance, especially considering that a large m may be necessary for the method's effectiveness.**
>
> Please refer to our response to Weakness 2, where we have conducted experiments on various values of $m$ and $n$.
>
> Continue with the next post

---

> ### Author Response · Authors · 2023-11-19
> **Reply to Reviewer XSqD (3/4)**
>
> **Q4: A discussion on the practical relevance of the derived upper bound in Theorem 3.5 and its connection to the actual classification task involving class labels would provide valuable insights.**
>
> Please check our response to Weakness 3.
>
> **Q5: The observation that the proposed approach performs significantly worse under certain settings, such as a random class prior matrix and non-constrained m, should be supported by clearer details on how test accuracies were computed, whether test data were balanced, and if class size variations in datasets played a role.**
>
> We computed the test accuracies on the test set of widely used benchmark datasets, including MNIST, Fashion, Kuzushiji, CIFAR-10, and SVHN. We simply used the given test set and did not make modifications to the class size of the provided test set.
>
> We also conducted additional experiments on the MNIST, Kuzushiji, and Fashion datasets with imbalanced classes. We constructed the class-imbalanced prior matrix by following a similar procedure to [1]. Firstly, we generate a random prior matrix $\Theta$. Secondly, we generate a geometric progression $\boldsymbol{l}$ according to the imbalance ratio $\rho$, where $\rho$ denotes the ratio between the largest and smallest elements in $\boldsymbol{l}$, i.e., $\boldsymbol{l}=\{1,\rho^{1/k-1}\,\rho^{2/k-1}\,\dots,\rho\}$. Thirdly, we divide each row of the prior matrix by this geometric progression, i.e., $\Theta_{ij}=\Theta_{ij}/$ $\boldsymbol{l}{j}$. Finally, we normalize each row of $\theta$ by $\Theta_{ij}=\Theta_{ij}/\sum_{j=1}^k\Theta_{ij}$ where $k$ is the class number.
>
> The experimental results are reported in Table 4. As can be seen from Table 4, our proposed methods RCM and CCM can still achieve satisfactory performance even under extreme class imbalance conditions. In MNIST and Kuzushiji datasets, RCM exhibited
> stronger robustness against class imbalance compared with CCM, as it was less affected as $\rho$ rises.
>
> Table 4: Classification performance of RCM and CCM trained on imbalanced classes.
>
> |   $\rho$    | Methods |      MNIST       |     Fashion      |    Kuzushiji    |
> | :---------: | :-----: | :--------------: | :--------------: | :-------------: |
> | 1 (balance) |   CCM   | 91.46$\pm$1.22%  | 76.09$\pm$07.57% | 72.85$\pm$2.52% |
> |             |   RCM   | **94.80$\pm$0.64%**  | **80.49$\pm$2.50%**  | **78.36$\pm$2.89%** |
> |     10      |   CCM   | 89.76$\pm$0.82%  | 73.21$\pm$7.96%  | 68.57$\pm$3.14% |
> |             |   RCM   | **93.27$\pm$1.05%**  | **75.36$\pm$3.53%**  | **74.03$\pm$2.67%** |
> |     100     |   CCM   | 72.49$\pm$14.81% | 64.77$\pm$2.46%  | 53.54$\pm$3.43% |
> |             |   RCM   | **85.32$\pm$2.75%**  | **65.43$\pm$5.26%**  | **62.69$\pm$2.31%** |
>
> [1] Learning Imbalanced Datasets with Label-Distribution-Aware Margin Loss, NeurIPS 2019.
>
> **Other suggestions.**
>
> We have incorporated the necessary changes according to your suggestions into our revised paper. Below we provide our responses to your suggestions.
>
> ***Clearly state whether data points are "independent from each other" or "dependent on each other for MCMU. Data points within each class are typically assumed to be independent, conditioned on their class labels. The statement in question uses "class priors" in place of "class labels", which could introduce confusion and should be clarified. Overall, given that label proportions and class priors are practically the same thing in the context studied by the papers it is hard to see any difference from a dependence perspective between the two approaches.***
>
> Thank you for your valuable suggestion! We have revised the description in our paper. In the LLP problem, data points are dependent on each other.
> The label proportions and class priors are two different notions in our paper, and we describe the differences between them in the following response.
>
> ***I think using the term "label proportions" when referring to class priors within each set would be more appropriate.***
>
> Thank you for your thoughtful consideration of our paper.
> In the MCMU problem, the unlabeled set is sampled according to class priors. In our humble opinion, we think that using class priors is more appropriate for MCMU. Although class priors and label proportions seem to be practically equivalent concepts, they actually describe two different sampling processes. In LLP, the notion of "label proportions" means that we calculate the proportion of instances belonging to each class after generating the data according to the underlying data distribution $p(x,y)$.
>
> Continue with the next post

---

> ### Author Response · Authors · 2023-11-19
> **Reply to Reviewer XSqD  (Part 4/4)**
>
> ***In Section 4.1. "The $n_i$ data points contained in $i$-th unlabeled set were randomly generated according to ..." Isn't this supposed to be "randomly sampled" or "drawn". Data already exists... nothing is supposed to be generated.***
>
> Thank you for pointing out this issue and we have corrected it in our paper.
>
> ***It would greatly enhance the paper if, within the related work section, an early explanation was provided regarding the reasons behind the possibility of negative empirical risk.***
>
> Thank you for this suggestion. There are two important reasons that we did not provide a detailed explanation/analysis of the negative empirical risk of previous work. Firstly, many previous studies on various weakly supervised learning problems have exhibited the negative empirical risk shared by common methods, such as positive-unlabeled learning [2], unlabeled-unlabeled learning [3], complementary-label learning [4], and so on. Thus we consider that this issue was commonly known and may not be necessary to be shown in this paper. Secondly, the page limit will also prevent us from adding a detailed explanation/analysis of the negative empirical risk of previous work. We really appreciate this suggestion and we will consider incorporating this part if our paper is accepted, because we are only allowed to use one more page for the camera-ready version.
>
> [2] Positive-unlabeled learning with non-negative risk estimator. NeurIPS 2017.
>
> [3] Mitigating Overfitting in Supervised Classification from Two Unlabeled Datasets: A Consistent Risk Correction Approach. AISTATS 2020.
>
> [4] Complementary-label learning for arbitrary losses and models. ICML 2019.
>
> ***On page 4 $i$ is used for indexing both the sets and data points. It may cause confusion.***
>
> Thank you for pointing out this issue and we have corrected it in our paper.
>
> ***In Theorem 3.6 the paper defines this probability $p(y=j|\overline{y},x)$ as the probability of $x$ in $\overline{y}$-th unlabeled set whose ground-truth label is $j$. Isn't this supposed to be the probability of $x$ in $\overline{y}$-th unlabeled set belonging to class $j$? $x$ may or may not belong to class $j$. $x$'s ground truth label is not necessarily $j$.***
>
> Yes, Your understanding is correct. $p(y=j|\overline{y},x)$ denotes the probability of $x$ in $\overline{y}$-th unlabeled set belonging to class $j$.
>
> ***Theorem 3.6 also considers $p(x|y=j)=p(x|y=j,\overline{y})$ as a fact. However, this is not necessarily true. The first one is the conditional distribution of class $j$ estimated using all data belonging to class $j$ whereas the second one is the conditional distribution of class $j$ estimated using only data from unlabeled set $\overline{y}$. So, from an empirical viewpoint, these distributions are not equal. Please clarify.***
>
> For the data generation of the MCMU problem, $p(x|y=j)=p(x|y=j,\overline{y})$ always holds. This is because, for MCMU, we need to first sample the information of true labels according to the class priors of each unlabeled set (i.e., $\overline{y} \rightarrow y$) and then sample the instances of each class according to the class-conditional density $p(x|y=j)$ (i.e., $y\rightarrow x$). Hence the whole data generation process is $\overline{y}\rightarrow y\rightarrow x$, which means, given the true label $y$, $x$ is independent of $\overline{y}$, i.e., $p(x|y=j)=p(x|y=j,\overline{y})$.
>
> ---------------------
>
> Finally, thank you again for your huge efforts in our paper. We sincerely hope that everything goes well with you!

---

> > ### Comment · Reviewer_XSqD · 2023-11-20
> >
> > Thanks for taking the time to do those extra comparisons I mentioned. It’s helpful to see how your method holds up with different 'm' values and varying levels of class imbalance. Based on these new results,  I'm happy to bump up my score by one.

---

> > > ### Author Response · Authors · 2023-11-20
> > >
> > > Thank you so much for increasing your score! We would like to express our highest respect to you for providing such a detailed and insightful review, which has definitely provided a great help for us to improve the quality of our paper!

---

### Official Review · Reviewer_S73G · 2023-10-29

**Soundness:** 4 excellent
**Presentation:** 3 good
**Contribution:** 4 excellent
**Rating:** 8
**Confidence:** 4

**Summary:**

This paper investigates an interesting weakly supervised learning problem called multi-class
classification from multiple unlabeled datasets, where only multiple sets of unlabeled data and their
class priors (i.e., the proportions of each class) are provided for training the classifier. To tackle this
problem, this paper first gives a multi-class extension of a previous work on binary classification from
multiple unlabeled datasets. However, this paper says that such a method still has several
disadvantages that limit the performance. So this paper further proposes a risk-consistent method that
can avoid those disadvantages and maintain theoretical guarantees. Experimental results support the
claim of this paper.

**Strengths:**

- Problem. This paper investigates the problem of multi-class classification from multiple unlabeled
datasets, which is interesting problem.
- Method. To solve the problem, this paper proposes two methods. The first one is a classifierconsistent method, which is a multi-class extension of a previous work on binary classification. The
second one is a risk-consistent method that can address the shortcomings of the first one.
- Theory. This paper gives theoretical analysis for the two methods proposed in this paper.
- Performance. From the experimental results, we can find that the classifier-consistent method can
achieve good performance compared with previous methods, and the risk-consistent method
outperforms all the methods. Some ablation studies also support the risk-consistent method.

**Weaknesses:**

- This paper has proposed two methods (CCM and RCM) to solve the problem and showed that the
second method is better than the first method. I think that there lacks a separate paragraph that is
specially for describing the difference between RCM and CCM in detail.
- This paper should give more explanations for the theoretical findings. For example, there are no
discussions or descriptions on Theorem 3.5 and Theorem 3.7.

**Questions:**

- What can we learn from Theorem 3.5 and Theorem 3.7?
- What is the relationship between the studied problem and the unlabeled-unlabeled learning problem
[Lu et al. (2019)]?

---

> ### Author Response · Authors · 2023-11-19
> **Reply to Reviewer S73G**
>
> Thank you for your valuable feedback. We sincerely appreciate your insightful comments.  Below is our response.
>
> **Weakness 1: This paper has proposed two methods (CCM and RCM) to solve the problem and showed that the second method is better than the first method. I think that there lacks a separate paragraph that is specially for describing the difference between RCM and CCM in detail.**
>
> Thank you for your valuable suggestion. We have adopted your suggestion to provide a separate paragraph to describe the distinctions between RCM and CCM in our revised paper.
>
> From a theoretical perspective, RCM could approximate or simulate the distribution of real clean data by utilizing the data distribution from the unlabeled set (by the importance-weighting schema). This means that RCM attempts to infer latent distribution patterns similar to those of real clean data from the unlabeled data. In contrast, CCM aims to better fit the distribution of the unlabeled set by maximizing a log-likelihood object. With a sufficient number of samples, RCM is more accurate in predicting the labels of unseen samples because it considers the restoration of the distribution of real clean data when modeling unlabeled data. This enables RCM to exhibit better generalization performance when facing unknown data, making more precise predictions for unseen samples.
>
> **Weakness 2: This paper should give more explanations for the theoretical findings. For example, there are no discussions or descriptions on Theorem 3.5 and Theorem 3.7.**
>
> Generally, the Rademacher complexity of the hypothesis space $H$ can be upper bounded by $C_{H}/\sqrt{n}$ for a positive constant $C_{H}$ [1,2]. Theorem 3.5 demonstrates that the empirical minimizer $\hat{f}_{\text{ccm}}$ would coverage to the minimizer $f^*$ on clean data as the training sample size approximates infinity. Theorem 3.7 shows that the empirical risk would converge to the expected risk as the sample size approaches infinity.
>
> [1] N. Golowich, A. Rakhlin, and O. Shamir. Size-independent sample complexity of neural networks. COLT 2018.
>
> [2] N. Lu, T.-Y. Zhang, G. Niu, and M. Sugiyama. Mitigating overfitting in supervised classification from two unlabeled datasets: A consistent risk correction approach. AISTATS, 2020.
>
> **Q1: What can we learn from Theorem 3.5 and Theorem 3.7?**
>
> Please check our response to Weakness 2.
>
> **Q2: What is the relationship between the studied problem and the unlabeled-unlabeled learning problem [Lu et al. (2019)]?**
>
> Both MCMU and the unlabeled-unlabeled learning aim to learn an instance-level classifier with the unlabeled sets. There are more constraints imposed on the problem setting of unlabeled-unlabeled learning,i.e., the number of classes is restricted to 2 (binary classification) and the number of unlabeled sets is also limited to 2. Our studied MCMU is more general and thus may have more potential applications.

---

### Official Review · Reviewer_3iq5 · 2023-11-06

**Soundness:** 3 good
**Presentation:** 3 good
**Contribution:** 4 excellent
**Rating:** 8
**Confidence:** 4

**Summary:**

This paper focuses on a newly proposed weakly supervised learning problem called multi-class classification from multiple unlabeled datasets (MCMU), where only multiple sets of unlabeled data and their class priors are provided in the training process. To solve this problem, this paper proposes two methods, including a classifier-consistent method (CCM) based on a probability transition function and a risk-consistent method (RCM) based on importance weighting. Additionally, theoretical analyses of the proposed methods and experimental results on multiple benchmark datasets are provided.

**Strengths:**

1. This paper studies a newly proposed weakly supervised learning problem called multi-class classification from multiple unlabeled datasets (MCMU), and proposes two effective methods, which could avoid the negative risk issue commonly encountered by previous unbiased risk estimator methods.
2. Theoretical analyses are provided to show the theoretical guarantees of the proposed methods.
3. Comprehensive experimental results on multiple benchmark datasets across various settings demonstrate the effectiveness of the proposed methods.
4. The paper is well organized and well written, which makes it easy to follow.

**Weaknesses:**

1. There is a lack of descriptions of Theorem 3.5 and Theorem 3.7. Additionally, could we compare RCM and CCM from a theoretical perspective?
2. The analysis in section 4.3 is interesting. However, there is a lack of discussion about the observation from Fig. 1. So, could the authors provide more details of why CCM is more robust when few data points are provided?
3. I noticed that in all experimental settings, the number of sets is greater than or equal to the number of classes. Could the authors provide a discussion where the number of sets is less than the number of classes?

**Questions:**

I have listed the questions in the weaknesses above. Please address them.

---

> ### Author Response · Authors · 2023-11-19
> **Reply to Reviewer 3iq5**
>
> Thank you for your constructive feedback. We really appreciate your insightful comments. Our responses to your concerns are provided as follows.
>
> **Q1: There is a lack of descriptions of Theorem 3.5 and Theorem 3.7.**
>
> Thank you for your advice, we have provided additional descriptions of Theorem 3.5 and Theorem 3.7 in our revised paper.
>
> Generally, the Rademacher complexity of the hypothesis space $H$ is upper bounded by $C_H/\sqrt{n}$ [1,2] for a constant $C_H$. Theorem 3.5 demonstrates that the empirical minimizer $\hat{f}_{\text{ccm}}$ would coverage to the minimizer $f^*$ on clean data as the training sample size approximates infinity. Theorem 3.7 shows that the empirical risk would converge to the expected risk as the sample size approaches infinity.
>
> [1] N. Golowich, A. Rakhlin, and O. Shamir. Size-independent sample complexity of neural networks. COLT 2017.
>
> [2] N. Lu, T.-Y. Zhang, G. Niu, and M. Sugiyama. Mitigating overfitting in supervised classification from two unlabeled datasets: A consistent risk correction approach. In AISTATS, 2020.
>
> **Additionally, could we compare RCM and CCM from a theoretical perspective?**
>
> From a theoretical perspective, RCM tries to approximate or simulate the distribution of real clean data by utilizing the data distribution from the unlabeled set (by the importance-weighting schema). This means that RCM attempts to infer latent distribution patterns of real clean data from the unlabeled data. In contrast, CCM aims to better fit the distribution of the unlabeled set by maximizing a log-likelihood object. With a sufficient number of samples, RCM is more accurate in predicting the labels of unseen samples because it considers the restoration of the distribution of real clean data when modeling unlabeled data. This enables RCM to exhibit better generalization performance when facing unknown data, making more precise predictions on unseen samples.
>
> **Q2: The analysis in section 4.3 is interesting. However, there is a lack of discussion about the observation from Fig. 1. So, could the authors provide more details of why CCM is more robust when few data points are provided?**
>
> We think it is because RCM is based on importance weighting, which requires a certain number of samples to accurately estimate the weights for reliably optimizing the objective. Hence, with a small sample size, CCM may outperform RCM.
>
> **Q3: I noticed that in all experimental settings, the number of sets is greater than or equal to the number of classes. Could the authors provide a discussion where the number of sets is less than the number of classes?**
>
> Thank you for your suggestion. We have additionally conducted experiments for the mentioned scenario where the number of sets is less than the number of classes, given a fixed number of data points.
>
> Table 1: Classification performance of each method on the MNIST dataset using a random prior matrix with variant set numbers. ($m \geq k$)
>
> | Methods | Unbiased        | U-Stop           | U-Correct       | U-Flood         | Prop            | U-PRR           | CCM             | RCM             |
> | ------- | --------------- | ---------------- | --------------- | --------------- | --------------- | --------------- | --------------- | --------------- |
> | m=5     | 18.23$\pm$8.08% | 45.88$\pm$12.29% | 34.71$\pm$7.20% | 58.97$\pm$4.25% | 72.48$\pm$2.37% | 26.34$\pm$6.46% | 88.80$\pm$2.43% | **89.16$\pm$4.29%** |
> | m=6     | 14.61$\pm$6.72% | 42.02$\pm$22.68% | 40.06$\pm$8.00% | 68.00$\pm$3.39% | 82.06$\pm$4.49% | 28.45$\pm$5.02% | 89.80$\pm$3.59% | **92.71$\pm$0.73%** |
> | m=7     | 15.56$\pm$2.38% | 42.39$\pm$6.74%  | 49.72$\pm$8.40% | 77.53$\pm$7.95% | 89.32$\pm$3.43% | 32.78$\pm$5.33% | 90.66$\pm$2.19% | **94.53$\pm$1.06%** |
> | m=8     | 15.13$\pm$0.87% | 37.08$\pm$16.58% | 32.65$\pm$2.24% | 79.89$\pm$4.45% | 88.57$\pm$3.11% | 27.72$\pm$0.64% | 89.99$\pm$2.75% | **94.49$\pm$1.06%** |
> | m=9     | 14.06$\pm$1.79% | 27.77$\pm$8.25%  | 33.80$\pm$1.88% | 82.83$\pm$2.78% | 88.31$\pm$3.68% | 29.79$\pm$4.22% | 91.42$\pm$1.26% | **94.44$\pm$0.91%** |
>
> The experimental results are reported in Table 1. In the experiments where $m$ is less than $k$, our proposed RCM and CCM consistently outperform other methods. Notably, RCM exhibits superior performance compared with CCM, and this performance difference becomes larger as $m$ increases.

---

> > ### Author Response · Authors · 2023-11-22
> > **Kind reminder to Reviewer 3iq5**
> >
> > Dear Reviewer 3iq5,
> >
> > Sorry to disturb you. We really appreciate that you have pointed out some issues and given constructive suggestions. Now we have addressed these issues and followed your suggestions to conduct additional experiments to show the case where the number of sets is less than the number of classes. Would you please kindly check that and consider re-evaluating our work? Please also let us know if you have any further concerns and we will be open to all possible discussions.

---

> > > ### Comment · Reviewer_3iq5 · 2023-11-22
> > > **I am happy to see your response, and I will raise the scores.**
> > >
> > > Thanks for the rebuttal. I am satisfied with it. I will update my score to credit the authors' efforts in extra experiments that demonstrate when the number of sets is small their method can still work well and showing this relaxation case can strengthen the work.

---

> > > > ### Author Response · Authors · 2023-11-22
> > > >
> > > > Thank you so much for raising your score! We really appreciate your endorsement to our work.

---

### Official Review · Reviewer_vMrL · 2023-11-06

**Soundness:** 3 good
**Presentation:** 1 poor
**Contribution:** 3 good
**Rating:** 6
**Confidence:** 3

**Summary:**

This paper proposes two algorithms (CCM and RCM) to classify multiple classes from multiple datasets using only class proportion information. They provide theoretical guarantees on the accuracy of their methods and show some experimental results.

**Strengths:**

I am not familiar with the related work. e.g. LLP, but the overall proposed method seems interesting and sufficiently novel. The proofs in the appendix seem reasonable and the additional experiments there exhaustive enough.

**Weaknesses:**

- See questions
-	There is typo / the sentence got broken up here: “where d is a positive integer denotes the input dimension. [k] = “
-	This is not grammatically correct / does not make sense “multiple-instance learning (Zhou et al., 2009)) could usually have access to weakly supervised labels.”

However, the paper is badly written (there is barely any explanation of anything; the appendix in some ways is more clear than the main paper!) and very difficult to understand. This is the biggest flaw of the paper, and makes it difficult to evaluate the paper accurately.

**Questions:**

-	The author’s state that Section 2.2 that in MCMU has the data generating process P(X|y) P(y|c) P(c) which seems reasonable, but that the generating process of LLP is P(y|X)? That would make sense as the modelling distribution (learning a discriminative function), but not as a data generating process? Also where is the class priors in the data generating process of LLP?
-	What are the class priors? The proportion of y_i for each of the k classes?
-	Section 2.4 is kind of randomly there without any introduction and it could be “cleaned up”.
-	What is WSL?
-	The proposed method is extremely unclear. The paper needs to explain more of its proposed approach instead of spending large parts of the text comparing to other papers e.g. Lu et al. and describing their approach as an extension / variation of other papers. (Note: this also unintentionally make it sound less novel)
-	The main idea of CCM seems to be that by converting the problem from classifying multiple classes in multiple datasets, it can be simplified into classifying which dataset the X sample comes from? And this equivalency is due to a deterministic transition function T() which is based on \rho the probability a datapoint belongs to a dataset, \theta the proportions of each class for each dataset, and \pi the proportion of each “class” over all datasets?
-	The problem setup seems very restrictive / artificial. There has to be the same number of classes for each dataset and the classes are “ordered” / aligned across multiple datasets? Also while the problem of classifying which dataset a sample belongs intuitively is “easier”, it is not clear how that solves the problem of which class within which dataset a sample belongs to. Multiplying by the proportions of dataset size and class size will “on average” get you probabilities on the latter problem, but this is just a math trick to achieve a bound. It is not clear how this works practically. The experiments seems to indicate it works, it is just extremely unintuitively how / explained badly.

---

> ### Author Response · Authors · 2023-11-19
> **Reply to Reviewer vMrL (Part 1/2)**
>
> Thank you for your valuable comments. We have corrected the identified typos in the manuscript. Below, we provide a detailed response to further enhance the clarity and quality of our work.
>
> **Q1: The author’s state that Section 2.2 that in MCMU has the data generating process P(X|y) P(y|c) P(c) which seems reasonable, but that the generating process of LLP is P(y|X)? That would make sense as the modelling distribution (learning a discriminative function), but not as a data generating process? Also where is the class priors in the data generating process of LLP?**
>
> The generation process in LLP can also be simplified by first generating a group of {$(x_{i},y_{i})$}$^n_{i=1}$ pairs randomly, and then calculating the corresponding class proportions based on the group of labels {$y_i$}$^n_{i=1}$. Thus the "class priors" in LLP are calculated by the group {$y_i$}$^n_{i=1}$ and we usually use the term "label proportions" (instead of "class priors") in LLP. In conclusion, for LLP, we first sample data pairs {$(x_i,y_i)$}$_{i=1}^n$ randomly according to probability density $p(x,y)$ then we could calculate the "label proportions".
>
> In contrast, for MCMU, we first set the "class priors", and generate data pairs {$(x_i,y_i)$}$_{i=1}^n$ according to the "class priors" as we described in Section 2.4.
>
> **Q2: What are the class priors? The proportion of $y_i$ for each of the $k$ classes?**
>
> In our paper, the class priors in MCMU denote the generative probabilities of classes, which means, we would like to generate training instances in each class according to the class priors. It is worth noting that the class priors used in MCMU are different from the label proportions used in LLP, where we calculate the proportion of instances belonging to each class after generating the data according to the underlying data distribution $p(x,y)$.
>
> **Q3: Section 2.4 is kind of randomly there without any introduction and it could be “cleaned up”.**
>
> Thanks for your suggestion. We have cleaned up it in Section 2.4.
>
> **Q4:  What is WSL?**
>
> In our paper, WSL refers to Weakly Supervised Learning. We have provided an explanation of WSL in the revised paper.
>
> **Q5: The proposed method is extremely unclear. The paper needs to explain more of its proposed approach instead of spending large parts of the text comparing to other papers e.g. Lu et al. and describing their approach as an extension / variation of other papers. (Note: this also unintentionally make it sound less novel)**
>
> Thanks so much for this wonderful suggestion. We have provided more detailed descriptions/explanations of our proposed methods in Appendix F.
>
> **Q6: The main idea of CCM seems to be that by converting the problem from classifying multiple classes in multiple datasets, it can be simplified into classifying which dataset the X sample comes from? And this equivalency is due to a deterministic transition function T() which is based on $\rho$ the probability a datapoint belongs to a dataset, $\theta$ the proportions of each class for each dataset, and $\pi$ the proportion of each "class" over all datasets?**
>
> The main idea of CCM is to design a loss function by converting the problem of classifying multiple classes into multiple unlabeled datasets.
> This conversion is based on our built connection relationship between $p(y|x)$ (the probability an instance $x$ belongs to a label) and $p(\overline{y}|x)$ (the probability an instance $x$ belongs to an unlabeled set) in Eq. (2), which is further represented as a transition function $T$ (i.e., $\overline{\eta}(x) = T(\eta(x))$ where $\overline{\eta}(x)=p(\overline{y}|x)$ and $\eta(x)=p(y|x)$).
> As we aim to approximate $p(\overline{y}|x)$ by $T(g(x))$, we can infer that $p(y|x)$ can be approximated by $g(x)$ (where we use $g(x)$ to denote the to represent the Softmax output of the model), because $T$ is an injective function. The detailed proof can be found in Appendix B.3.
> $\rho$ is the probability that a data point belongs to an unlabeled set, $\theta$ represents the priors of each class for each unlabeled set, and $\pi$ is the proportion of each "class" overall unlabeled sets.
>
> Continue with the next post

---

> ### Author Response · Authors · 2023-11-19
> **Reply to Reviewer vMrL (Part 2/2)**
>
> **Q7: The problem setup seems very restrictive / artificial. There has to be the same number of classes for each dataset and the classes are “ordered” / aligned across multiple datasets? Also while the problem of classifying which dataset a sample belongs intuitively is “easier”, it is not clear how that solves the problem of which class within which dataset a sample belongs to. Multiplying by the proportions of dataset size and class size will “on average” get you probabilities on the latter problem, but this is just a math trick to achieve a bound. It is not clear how this works practically. The experiments seem to indicate it works, it is just extremely unintuitively how / explained badly.**
>
> We would like to clarify that our setting does not have the constraint that the number of classes in each unlabeled set must be the same. There could be different numbers of classes in different unlabeled sets. For example, if there is no class $j$ exists in unlabeled set $i$. We can simply set $\theta_{ij}$ (the $j$-th class prior of the $i$-th unlabeled set) to 0, then we can still handle this setting by our proposed methods.
>
> We admit that the classes should be ordered/aligned across multiple unlabeled sets. This is actually the basic requirement to learn a single classifier from multiple datasets simultaneously. Besides, in our humble opinion, it is also easy to satisfy this requirement during the data collection process, by simply assigning each collected class label a unique index.
>
> In our humble opinion, the MCMU problem is not artificial but can be encountered in various real-world scenarios. For example, predicting the demographic information of users in social networks is crucial for practical policy-making [1]. However, the collection of individual user information might be constrained by data privacy concerns. Fortunately, acquiring multiple unlabeled datasets is more feasible, and their class priors can be derived from existing census data.
>
> [1] Culotta A, Kumar N, Cutler J. Predicting the demographics of twitter users from website traffic data. AAAI 2015.
>
> **Q8: Solves the problem of which class within which dataset a sample belongs to.**
>
> Please check our response to Q6.

---

> > ### Author Response · Authors · 2023-11-21
> > **Kind reminder to Reviewer vMrL**
> >
> > Dear Reviewer vMrL:
> >
> > Sorry to disturb you. We sincerely appreciate your valuable comments. We understand that you may be too busy to check our rebuttal. We would like to further provide brief answers here to the issues that may be your primary concerns.
> >
> > We really appreciate your efforts in identifying the typos and ambiguous descriptions, as well as providing valuable suggestions for the problem setting and method descriptions. In response to your feedback, we have carefully revised the typos and enhanced the clarity of our descriptions regarding the problem setting and the methods we proposed. We have provided a specific description of the technique details of our proposed methods in Appendix F.
> >
> > We guess that you may have some concerns regarding how CCM addresses the assignment of an example to a specific class. As we aim to approximate $p(\overline{y}|x)$ by $T(g(x))$, we can infer that $p(y|x)$ can be approximated by $g(x)$ (where we use $g(x)$ to denote the Softmax output of the model) since $T$ is an injective function. The detailed proof can be found in Appendix B.3.
> >
> > In addition, we would like to clarify that our methods do not require each unlabeled set to contain the same number of classes. This issue can be effectively resolved by appropriately setting the values of the class prior matrix.
> >
> > We sincerely hope that the above answers can address your concerns. We look forward to your response and are willing to answer any questions.

---

> > > ### Comment · Reviewer_vMrL · 2023-12-04
> > >
> > > I have re-read the revised paper and comments and updated my score accordingly. I remain unconvinced that in real life settings the multiple unlabeled datasets will be completely identically distributed, which is what the experimental setting is showing by breaking separating one real dataset into multiple unlabeled datasets.

---

### Official Review · Reviewer_x6yh · 2023-11-07

**Soundness:** 2 fair
**Presentation:** 3 good
**Contribution:** 2 fair
**Rating:** 6
**Confidence:** 5

**Summary:**

The authors study the challenge of weakly supervised learning with a focus on multi-class classification from multiple unlabeled datasets. They propose two methods: the Classifier-Consistent Method (CCM), which utilizes class priors and a probability transition function for training, and the Risk-Consistent Method (RCM), which aims to enhance the CCM by ensuring risk consistency through importance weighting to refine supervision during training. The study claims the superiority of these methods, supporting them with comprehensive theoretical analyses for statistical consistency and positive experimental results across multiple benchmark datasets.

**Strengths:**

1. The paper presents a novel approach based on statistical learning theory to solve the problem of multi-class classification from multiple unlabeled datasets and the theoretical guarantees of the estimation error bounds strengthen the claims regarding its effectiveness.
2. The authors provide a clear and comprehensive description of their methodology, which enables other researchers to reproduce their results.

**Weaknesses:**

1. The methods and the corresponding theory presented in the paper are sound and relevant, but the lack of sufficient originality and novelty compared to a previously published work [1,2] could be a limitation. Therefore, the authors should carefully consider these points and take the necessary steps to distinguish their work and demonstrate its original contributions.

   [1] Feng, L., Lv, J., Han, B., Xu, M., Niu, G., Geng, X., ... & Sugiyama, M. (2020). Provably consistent partial-label learning. Advances in neural information processing systems, 33, 10948-10960.

   [2] Kobayashi, R., Mukuta, Y., & Harada, T. (2022). Learning from Label Proportions with Instance-wise Consistency. *arXiv preprint arXiv:2203.12836*.

2. Employing the direct outputs of a network to estimate the posterior probabilities $p(y=j∣x)$ can be imprecise, particularly under the extreme weak supervision scenario where only class prior probabilities serve as supervisory signals. In such a setting, the network predictions may not align well with the true posterior distributions, leading to suboptimal performance.

3. In Theorems 3.5 and 3.7, the assumptions regarding the Lipschitz continuity of the loss function present an indeterminable strength which could affect the assessment of the model's robustness. Furthermore, the generalization error bound incorporates a summation over $k$ categories of the Rademacher complexity, which may result in a rather loose bound.

4. The use of subscripts $i$ and $j$ appears to be somewhat confusing and may lead to difficulty in understanding for the reader. Specifically, the subscripts switch between $i$ and $j$, which could introduce ambiguity in distinguishing between the elements associated with the k-dimensional vector $\eta(x)$ and the m-dimensional vector $\bar{\eta}(x)$. Clarity in the mathematical notation is crucial for the precise communication of such theoretical results.

**Questions:**

Seen weaknesses.

---

> ### Author Response · Authors · 2023-11-19
> **Reply to Reviewer x6yh (Part 1/2)**
>
> Thanks for the reviewer's valuable comments. Our point-by-point responses are provided as follows.
>
> **Q1: The methods and the corresponding theory presented in the paper are sound and relevant, but the lack of sufficient originality and novelty compared to a previously published work [1,2] could be a limitation. Therefore, the authors should carefully consider these points and take the necessary steps to distinguish their work and demonstrate its original contributions.**
>
> We would like to admit that our proposed RCM shares a similar high-level idea, i.e., the importance-weighting (IW) strategy, compared with [1]. However, in our humble opinion, the originality/novelty of our work is sufficient, because of the following reasons:
>
> 1) This is the first time that the IW strategy is used in the MCMU problem. It is worth noting that the IW strategy has been widely used in various weakly supervised learning problems, including Partial Label Learning [1], Noisy Label Learning [3,4,5], Learning from Label Proportions [2] (which is different from our studied MCMU, as explained in our paper), and so on. Our paper provides the first attempt to apply the IW strategy to the MCMU problem. In addition, the computational complexity of the method in [2] gets significantly larger as the size of unlabeled sets increases. Therefore, [2] introduces an approximation method, however, this approximation method does not have theoretical guarantees (i.e., the risk equivalence). In contrast, the computational complexity of our method will not be influenced by the increased size of unlabeled sets, and thus our method does not require approximation, hence our method always holds the risk equivalence.
>
> 2) It is not straightforward to derive our risk of MCMU that is equivalent to the fully supervised classification risk. This is because we need to consider some special properties of the MCMU problem to derive the desired risk. Hence, the derivation process can also reflect the originality/novelty of our work. Concretely, compared with [1], applying the IW strategy to the MCMU problem has the following difficulties. (a) In Partial Label Learning, the partial label $Z$ is assigned to a single instance, while in MCMU, the class priors $\theta$ is assigned to a group of instance. (b) The probability dependency between the true label $y$ and the class priors $\theta$ is much more complex compared with the dependency between the true label $y$ and the partial label $Z$. $p(y|\theta)$ is a real number between 0 and 1, while $p(Z|y)$ takes value 0 when $y\in Z$ or take the value $1/(2^{k-1}-1)$ when $y\notin Z$.
>
> **Q2: Employing the direct outputs of a network to estimate the posterior probabilities $p(y|x)$ can be imprecise, particularly under the extreme weak supervision scenario where only class prior probabilities serve as supervisory signals. In such a setting, the network predictions may not align well with the true posterior distributions, leading to suboptimal performance.**
>
> We agree with the reviewer to some degree that the Softmax outputs may not precisely estimate the posterior probabilities. However, from the theoretical perspective, our provided Lemma 3.3 demonstrates that if certain loss functions are used (e.g., cross-entropy loss), the Softmax outputs of the model can ideally approximate the true posterior distributions. This theoretical result motivates us to use the Softmax outputs to estimate the posterior probabilities.
>
> From the empirical perspective, it is true that the Softmax outputs may not precisely estimate the posterior probabilities. However, there are still many previous methods that have applied the Softmax outputs to approximate the posterior probabilities [1,3,4,5] and achieved excellent performance on various tasks. Hence, the rationale for employing Softmax outputs to approximate posterior probabilities has been widely acknowledged in previous studies. Actually, how to precisely estimate the posterior probabilities is another important task called model calibration [6], which is definitely out of the scope of this paper. In the current stage, we cannot expect there exists a perfect estimation strategy that has no estimation error. That is why we simply used the Softmax outputs to estimate the posterior probabilities, in our paper.
>
> Continue with the next post.

---

> ### Author Response · Authors · 2023-11-19
> **Reply to Reviewer x6yh (Part 2/2)**
>
> **Q3: In Theorems 3.5 and 3.7, the assumptions regarding the Lipschitz continuity of the loss function present an indeterminable strength which could affect the assessment of the model's robustness. Furthermore, the generalization error bound incorporates a summation over $k$ categories of the Rademacher complexity, which may result in a rather loose bound.**
>
> The Lipschitz continuity assumption was widely accepted in the machine learning community [1,3,4,5] for establishing generalization or estimation error bounds. This property has been chosen for its stability-enhancing characteristics in optimization and theoretical analyses, contributing to the theoretical underpinning of our work. Importantly, our use of Lipschitz continuity does not compromise the robustness assessment of the model under investigation.
>
> Furthermore, it is noteworthy that Lipschitz continuity has become a fundamental property of many loss functions, such as cross-entropy loss, mean absolute error, square loss, and so on. The widespread applications of these loss functions have demonstrated the rationality of Lipschitz continuity.
>
> For the generalization error bound and the summation over $k$ categories of the Rademacher complexity. The summation of $k$ Rademacher complexities is theoretically supported by the Rademacher vector contraction inequality [7], with the fact that MCMU is a $k$-class classification problem. According to [7], the summation of $k$ Rademacher complexities will not result in a loose bound.
> It is also worth noting that similar bounds have been established in many previous studies on weakly supervised learning, e.g., [1,3,5].
>
> **Q4: The use of subscripts $i$ and $j$ appears to be somewhat confusing and may lead to difficulty in understanding for the reader. Specifically, the subscripts switch between $i$ and $j$, which could introduce ambiguity in distinguishing between the elements associated with the k-dimensional vector $\eta(x)$ and the m-dimensional vector $\overline{\eta}(x)$. Clarity in the mathematical notation is crucial for the precise communication of such theoretical results.**
>
> Thank you very much for your valuable suggestions. We have made some necessary revisions accordingly, to improve the clarity of mathematical notations in our paper.
>
> **References:**
>
> [1] Feng L, Lv J, Han B, et al. Provably consistent partial-label learning. NeurIPS 2020.
>
> [2] Kobayashi R, Mukuta Y, Harada T. Learning from Label Proportions with Instance-wise Consistency. arXiv preprint arXiv:2203.12836, 2022.
>
> [3] Wu S, Xia X, Liu T, et al. Class2simi: A noise reduction perspective on learning with noisy labels. ICML 2021.
>
> [4] Liu T, Tao D. Classification with noisy labels by importance reweighting. TPAMI 2016.
>
> [5] Xia X, Liu T, Wang N, et al. Are anchor points really indispensable in label-noise learning? NeurIPS 2019.
>
> [6] Guo C, Pleiss G, Sun Y, et al. On calibration of modern neural networks. ICML 2017.
>
> [7] Zatarain-Vera O. A vector-contraction inequality for Rademacher complexities using $p$-stable variables. ALT 2016.

---

> > ### Comment · Reviewer_x6yh · 2023-11-20
> >
> > Thank you for your comprehensive response. Your clarifications and revisions have addressed all of my concerns.  I have decided to increase my rating.

---

> > > ### Author Response · Authors · 2023-11-20
> > >
> > > Thank you so much for the reply! We are so glad to see that we have addressed your concerns. We sincerely thank you for acknowledging our contributions!

---

### Meta-Review · Area_Chair_mMHT · 2023-12-09

**Metareview:**

This paper was reviewed by five experts and received 6, 6, 8, 8, 6 as the final ratings. It was extensively discussed between the authors and the reviewers during the rebuttal period. The reviewers concurred that the paper proposes a novel approach for multi-class classification from multiple unlabeled datasets; the proposed method is theoretically sound and provides guarantees on the error bounds; comprehensive experimental studies have been conducted on multiple benchmark datasets. The reviewers raised a few questions about the problem setup, writing clarity and assumptions regarding the Lipschitz continuity of the loss function, which were all convincingly addressed by the authors in the rebuttal. We particularly appreciate the authors’ efforts in conducting additional experiments on the MNIST dataset to address the concern about the performance of the method, when the number of sets is less than the number of classes, raised by Reviewer 3iq5. We also appreciate the authors’ efforts in conducting additional experiments on three datasets (MNIST, Fashion and Kuzushiji) to address the concerns regarding the performance of the method under class imbalance settings, raised by Reviewer XSqD.

The reviewers, in general, have a positive opinion about the paper and its contributions. All of them have been satisfied with the authors’ rebuttal and have recommended acceptance. Based on the reviewers’ feedback, the decision is to recommend the paper for acceptance to ICLR 2024. We congratulate the authors on the acceptance of their paper!

**Justification For Why Not Higher Score:**

One of the reviewers was unconvinced that in real life settings, multiple unlabeled datasets will be completely identically distributed; this concern was not convincingly addressed by the authors in the rebuttal.

**Justification For Why Not Lower Score:**

The reviewers all agreed that the paper addresses a novel problem, is theoretically sound, and the conducts an exhaustive set of experimental studies. The author rebuttal has addressed most of the concerns raised by the reviewers. This paper will thus be a good addition to the list of spotlight presentations at ICLR 2024.

---

### Decision · Program_Chairs · 2024-01-16

Accept (spotlight)